# In vivo generation of bone marrow from embryonic stem cells in interspecies chimeras

**Bingqiang Wen[1], Guolun Wang[1], Enhong Li[1], Olena A Kolesnichenko[1], Zhaowei Tu[2], Senad Divanovic[3,4], Tanya V Kalin[4,5], Vladimir V Kalinichenko[1,4,5,6]\***

[1]Center for Lung Regenerative Medicine, Perinatal Institute, Cincinnati Children's Hospital Medical Center, Cincinnati, United States; [2]Division of Experimental Hematology and Cancer Biology, Cincinnati Children's Hospital Medical Center, Cincinnati, United States; [3]Division of Immunobiology, Cincinnati Children's Hospital Medical Center, Cincinnati, United States; [4]Department of Pediatrics, College of Medicine of the University of Cincinnati, Cincinnati, United States; [5]Division of Pulmonary Biology, Cincinnati Children's Hospital Medical Center, Cincinnati, United States; [6]Division of Developmental Biology, Cincinnati Children's Hospital Medical Center, Cincinnati, United States

**\*For correspondence:**
Vladimir.Kalinichenko@cchmc.org

**Competing interest:** The authors declare that no competing interests exist.

**Abstract** Generation of bone marrow (BM) from embryonic stem cells (ESCs) promises to accelerate the development of future cell therapies for life-threatening disorders. However, such approach is limited by technical challenges to produce a mixture of functional BM progenitor cells able to replace all hematopoietic cell lineages. Herein, we used blastocyst complementation to simultaneously produce BM cell lineages from mouse ESCs in a rat. Based on fluorescence-activated cell sorting analysis and single-cell RNA sequencing, mouse ESCs differentiated into multiple hematopoietic and stromal cell types that were indistinguishable from normal mouse BM cells based on gene expression signatures and cell surface markers. Receptor–ligand interactions identified *Cxcl12-Cxcr4*, *Lama2-Itga6*, *App-Itga6*, *Comp-Cd47*, *Col1a1-Cd44*, and *App-Il18rap* as major signaling pathways between hematopoietic progenitors and stromal cells. Multiple hematopoietic progenitors, including hematopoietic stem cells (HSCs) in mouse–rat chimeras derived more efficiently from mouse ESCs, whereas chondrocytes predominantly derived from rat cells. In the dorsal aorta and fetal liver of mouse–rat chimeras, mouse HSCs emerged and expanded faster compared to endogenous rat cells. Sequential BM transplantation of ESC-derived cells from mouse–rat chimeras rescued lethally irradiated syngeneic mice and demonstrated long-term reconstitution potential of donor HSCs. Altogether, a fully functional BM was generated from mouse ESCs using rat embryos as 'bioreactors'.

## Editor's evaluation

This work convincingly establishes a chimeric blastocyst complementation assay as a "bioreactor" to study the differentiation of mouse embryonic stem cells into hematopoietic lineages. The elegance of the approach lies in the use of GFP+ mouse embryonic stem cells that are implanted into a rat blastocyst, thus allowing for the tracking and phenotyping of the mouse-derived GFP+ hematopoietic cells in the post-natal rat. This is an important contribution that will be of interest to researchers in developmental biology and hematopoiesis.

## Introduction

The bone marrow (BM) is a remarkably complex organ consisting of multiple mesenchymal, immune, endothelial, and neuronal cell types which together comprise a highly specialized microenvironment required to support c blood regeneration or hematopoiesis (*Baccin et al., 2020*; *Baryawno et al., 2019*; *Rowe et al., 2016*; *Tikhonova et al., 2019*; *Vo and Daley, 2015*). Hematopoiesis occurs in a stepwise manner and is initiated by a heterogeneous, multipotent, population of hematopoietic stem cells (HSCs), located at the apex of the hematopoietic differentiation tree. Long-term HSCs (LT-HSCs) remain quiescent to maintain their undifferentiated state within the BM niche. When necessary, LT-HSCs can either undergo differentiation or self-renewal, to maintain the HSC pool. Conversely, short-term HSCs (ST-HSCs) are restricted in their self-renewal capacity and primed for differentiation into multipotent progenitors (MPPs), initiating the process of blood cell development. MPPs further differentiate into common myeloid progenitors (CMPs), lymphoid-primed multipotent progenitors (LMPPs), and common lymphoid progenitors (CLPs) that become increasingly lineage restricted with subsequent cell divisions, ultimately yielding all mature blood cell types (*Haas et al., 2018*). The complexities of the hematopoietic system have been studied extensively in vitro, utilizing paired-daughter and colony-forming unit (CFU) assays (*Rowe et al., 2016*; *Vo and Daley, 2015*). Fluorescence-activated cell sorting (FACS) has allowed for precise isolation and characterization of HSCs and progenitor populations based on cell surface markers. Classically, the most biologically relevant way to test HSC function remains to be through serial transplantation and hematopoietic reconstitution of irradiated recipient mice (*Purton and Scadden, 2007*; *Rowe et al., 2016*; *Vo and Daley, 2015*). Recent advances in single-cell RNA sequencing (scRNAseq) have made it possible to further explore heterogeneity of the BM niche (*Baryawno et al., 2019*; *Tikhonova et al., 2019*), and identify gene expression signatures of hematopoietic progenitor cells as they differentiate into mature blood cell types (*Baccin et al., 2020*; *Nestorowa et al., 2016*).

Generation of functional BM from embryonic stem cells (ESCs) or induced pluripotent stem cells (iPSCs) promises to provide new therapeutic opportunities for hematologic and autoimmune disorders. However, this approach is limited by technical challenges to produce functional HSCs or the mixture of hematopoietic progenitors capable of replacing all mature blood cell types after cell transplantation. HSC-like cells have been generated from mouse and human ESCs and iPSCs using in vitro differentiation protocols (*Amabile et al., 2013*; *Doulatov et al., 2013*; *Grigoriadis et al., 2010*; *Kitajima et al., 2011*; *Ledran et al., 2008*; *Sugimura et al., 2017*; *Vodyanik et al., 2006*). Likewise, ESCs and iPSCs have been used to produce myeloid and lymphoid progenitor cells as well as differentiated hematopoietic cells, including neutrophils, monocytes, erythroid cells, and T and B lymphocytes (*Doulatov et al., 2013*; *Elcheva et al., 2014*; *Galic et al., 2006*; *Kennedy et al., 2012*; *Montel-Hagen et al., 2019*; *Nafria et al., 2020*; *Vodyanik et al., 2005*). When transplanted into irradiated animals, ESC/iPSC-derived hematopoietic progenitor cells undergo differentiation and engraft into the BM niche, providing an important source of renewal and regeneration for various blood cell lineages (*Rowe et al., 2016*; *Sugimura et al., 2017*; *Vo and Daley, 2015*). While ESC/iPSC-derived hematopoietic cells often express appropriate cell markers, gene expression and functional studies indicate significant differences between ESC/iPSC-derived cells and endogenous cells that have undergone normal morphogenesis in the BM (*Lin et al., 2019*; *Lu et al., 2016*; *Sugimura et al., 2017*).

In vivo differentiation of ESCs into multiple cell lineages can be achieved using blastocyst complementation, in which donor ESCs are injected into blastocysts of recipient animals to create chimeras. Fluorescently labeled ESCs undergo differentiation in recipient embryos that serve as 'biological reactors' by providing growth factors, hormones, and cellular niches to support ESC differentiation in the embryo. In mouse and rat apancreatic Pdx1⁻/⁻ embryos, donor ESCs formed an entire pancreas in which both exocrine and endocrine cells were almost entirely derived from ESCs or iPSCs (*Kobayashi et al., 2010*; *Yamaguchi et al., 2017*). Mouse ESC/iPSC-derived β-cells from mouse–rat chimeras were fully differentiated and successfully rescued syngeneic diabetic mice (*Yamaguchi et al., 2017*). ESCs generated pancreatic cell lineages in apancreatic pigs (*Matsunari et al., 2013*), kidney in Sall1-deficient rats (*Goto et al., 2019*), endothelial cells in Flk1⁻/⁻ mice (*Hamanaka et al., 2018*), lymphocytes in immunodeficient mice (*Muthusamy et al., 2011*), and neuronal progenitors in mice with forebrain-specific overexpression of diphtheria toxin (*Chang et al., 2018*). Recently, mouse ESCs were used to generate lung and thyroid tissues in embryos deficient for Fgf10, Nkx2-1, Fgfr2, or β-catenin (*Kitahara et al., 2020*; *Mori et al., 2019*; *Wen et al., 2021*). Using blastocyst complementation,

mouse ESCs effectively produced hematopoietic cells in mice deficient for Kit or Flk1 (*Hamanaka et al., 2018*; *Jansson and Larsson, 2010*). ESC-derived endothelial progenitor cells from mouse–rat chimeras were indistinguishable from endogenous endothelial progenitor cells based on gene expression signatures and functional properties (*Wang et al., 2021*), indicating that ESC/iPSC-derived progenitors can be used for tissue regeneration (*Bolte et al., 2020a*; *Bolte et al., 2018*; *Dharmadhikari et al., 2015*; *Kolesnichenko et al., 2021*). While all these studies support the effectiveness of blastocyst complementation for differentiation of multiple cell types from ESCs/iPSCs in vivo, generation of functional BM from ESCs in interspecies chimeras has not yet been achieved.

Herein, we used blastocyst complementation to produce mouse BM in a rat. ESC-derived cells from multiple hematopoietic and stromal cell lineages were indistinguishable from normal mouse BM cells based on gene expression signatures and cell surface markers. Transplantation of ESC-derived BM cells into lethally irradiated syngeneic mice prevented mortality and resulted in a long-term contribution to BM and mature blood cell types. Our data demonstrate that interspecies chimeras can be used as 'bioreactors' for in vivo differentiation and functional studies of ESC-derived BM hematopoietic and stromal cells.

## Results
### Generation of BM from pluripotent ESCs in interspecies mouse–rat chimeras
To determine whether mouse ESCs can differentiate into multiple hematopoietic cell lineages in the BM of a rat, blastocyst complementation was performed by injecting GFP-labeled mouse C57BL/6 ESCs (ESC-GFP) into rat SD blastocysts to create interspecies mouse–rat chimeras. Chimeric embryos were transferred into surrogate female rats for subsequent development in utero (*Figure 1A*). While mouse–rat chimeras were viable, they were smaller than age-matched rats (*Figure 1B*). Consistent with the presence of mouse ESC-derived cells (black) in the skin tissue (*Wang et al., 2021*), mixed black and white pigmentation distinguished the mouse–rat chimeras from juvenile rats (*Figure 1B*). The average body weight of mouse–rat chimeras was smaller than rats, but larger than mice of similar age (*Figure 1C*). ESC-derived cells were abundant in femur and tibia bones of the chimeras as evidenced by GFP fluorescence (*Figure 1D*). FACS analysis of BM cells obtained from juvenile mouse–rat chimeras revealed that the percentage of ESC-derived cells was 15–50% (*Figure 1E, F*). Thus, ESCs contribute to the BM of mouse–rat chimeras.

To identify ESC-derived HSCs, we used GFP fluorescence and mouse-specific antibodies recognizing multiple cell surface antigens (*Figure 1E* and *Figure 1—figure supplement 1A, B*). First, ESC-derived GFP$^+$ BM cells were subdivided into *lineage-positive* (Lin$^+$) and *lineage-negative* subpopulations (Lin$^-$) (*Figure 1E* and *Figure 1—figure supplement 1A, B*). The percentage of ESC-derived Lin$^-$ cells in the BM of mouse–rat chimeras was similar to the percentage of Lin$^-$ cells in the BM of age-matched C57BL/6 mice (*Figure 1E, G*). Next, we used Sca1 and CD117 (c-KIT) antibodies to identify Lin$^-$Sca1$^+$c-KIT$^+$ cells (LSKs) (*Figure 1E*). The percentage of LSKs was higher in the BM of mouse–rat chimeras compared to the control (*Figure 1G*). Based on cell surface expression of CD150 and CD48, the percentage of LT-HSCs among LSKs was also higher in mouse–rat chimeras (*Figure 1E, H*). While changes in ST-HSCs were not significant (*Figure 1H*), total numbers of HSCs (LT-HSCs + ST-HSCs) were higher in mouse–rat chimeras compared to mice of the same age (*Figure 1I*). Thus, mouse ESCs can differentiate into hematopoietic progenitor cells in the BM of mouse–rat chimeras.

### Single-cell RNA sequencing identifies multiple subpopulations of ESC-derived hematopoietic cells in the BM of mouse–rat chimeras
To identify ESC-derived cells in the BM, single-cell RNAseq (the 10× Chromium platform) of FACS-sorted GFP$^+$ BM cells was performed. Mouse ESC-derived cells from P10 mouse–rat chimeras were compared to ESC-derived cells from P10 mouse–mouse (control) chimeras, the latter of which were produced by complementing mouse blastocysts with mouse ESCs from the same ESC-GFP cell line. Based on GFP fluorescence, contribution of ESCs to BM cells in both chimeras was similar (*Figure 2—figure supplement 1A, B*). Since the numbers of HSCs and other hematopoietic progenitor cells in the BM are low compared to numbers of differentiated hematopoietic cells, we enriched for BM progenitor cell populations prior to single-cell RNA sequencing by combining 90% of FACS-sorted

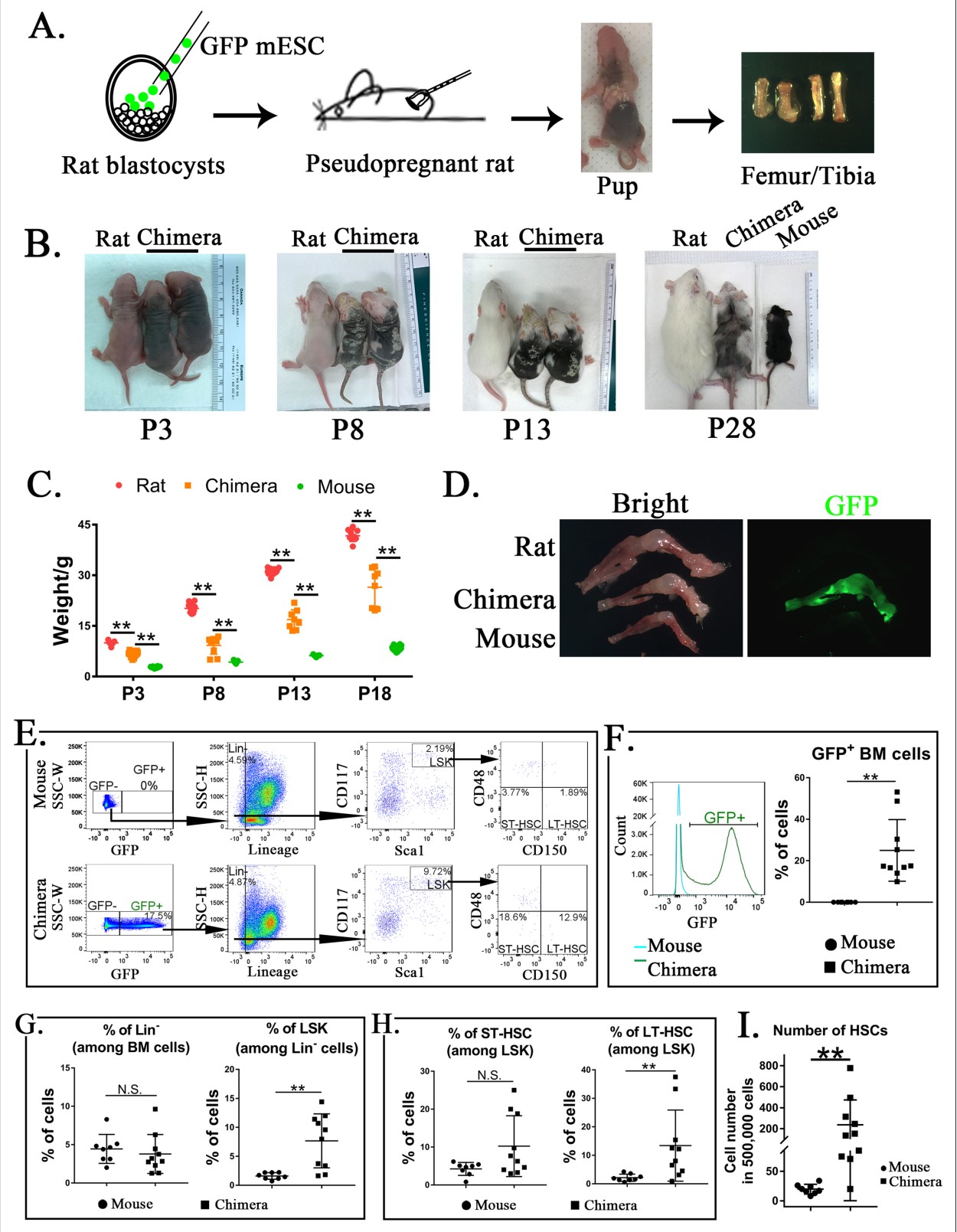

**Figure 1.** Mouse embryonic stem cells (ESCs) contribute to hematopoietic stem cells (HSCs) in the bone marrow (BM) of mouse–rat chimeras. (**A**) Schematic shows blastocyst complementation of rat embryos with mouse ESCs to generate interspecies mouse–rat chimeras. GFP-labeled mouse ESCs (mESCs) were injected into rat blastocysts, which were implanted into surrogate rat females to undergo embryonic development in utero. Femur and tibia bones of the chimeras were used to obtain BM cells. (**B**) Photographs of mouse–rat chimeras are taken at postnatal (P) days P3, P8, P13, and P28.

*Figure 1 continued on next page*

*Figure 1 continued*

Mixed black and white pigmentation distinguishes the mouse–rat chimeras from juvenile rats and mice. (**C**) Weights of mouse–rat chimeras are shown at different time points and compared to rats and mice of similar ages. Chimeras are significantly smaller than rats, but larger than mice (*n* = 7–18 in each group), **p < 0.01, see also *Source data 1*. (**D**) Fluorescence microscopy shows GFP and bright-field images of femur and tibia bones from P4 rat, mouse, and mouse–rat chimera. (**E**) Fluorescence-activated cell sorting (FACS) analysis of mouse ESC-derived (GFP-positive) cells in the BM of P10 mouse–rat chimeras. Lineage-negative (Lin⁻), LSK, short-term HSC (ST-HSC), and long-term HSC (LT-HSC) cell subsets were identified in the BM of mouse–rat chimeras (*n* = 10) and control mice (*n* = 8), see also *Figure 1—figure supplement 1A, B*. (**F**) Histograms show GFP fluorescence of BM cells from chimeras and control mice. (**G–H**) FACS analysis shows increased percentages of mouse LSKs and LT-HSCs in BM of mouse–rat chimeras (*n* = 10) compared to control mice (*n* = 8), **p < 0.01, N.S. indicates no significance. (**I**) FACS analysis shows increased numbers of HSCs (ST-HSCs + LT-HSCs) in BM of mouse rat chimeras (*n* = 10) compared to control mice (*n* = 8), **p < 0.01.

The online version of this article includes the following figure supplement(s) for figure 1:

**Figure supplement 1.** Identification of lineage⁻ cells, LSKs, short-term HSCs (ST-HSCs), and long-term HSCs (LT-HSCs) in the bone marrow (BM).

GFP⁺Lin⁻ cells and 10% of GFP⁺Lin⁺ cells in each experimental group. BM cells from 3 animals per group were combined prior to FACS sorting. Based on published gene expression signatures of mouse BM cells (*Baccin et al., 2020*), 11,326 cells from 14 major cell subtypes were identified: 5308 cells from control mouse–mouse chimeras and 6018 cells from mouse–rat chimeras. These include lymphoid, erythroid, myeloid, and neutrophil progenitors, Pro-B, Pre-B, B and T lymphocytes, mega-karyocytes, dendritic cells, neutrophils, basophils/eosinophils, monocytes, and LMPP cells (*Figure 2A* and *Figure 2—figure supplement 2A*). Analysis of BM cells from mouse–rat and mouse–mouse chimeras demonstrated similar distributions of hematopoietic cell lineages derived from CMP and CLP (*Figure 2A*), indicating identical cell types in mouse–rat and control chimeras. For selected genes, we used violin plots to confirm cell specificity and expression levels of *Ptprc (Cd45)*, *Pclaf*, *Vpreb1*, *Tmpo*, *Ebf1*, *Ms4a4b*, *Vamp5*, *Elof1*, *Elane*, *Ms4a2*, *Siglech*, *Ngp*, *Clec4d*, *Ctss*, and *Ftl1-ps1* in the combined dataset (*Figure 2—figure supplement 3*). Markers of endothelial cells, adipocytes, osteo-cytes, and neuronal cells were undetectable in BM cell suspensions from both chimeras (*Figure 2—figure supplement 2B*). Percentages CLP-derived lymphoid progenitors, Pro-B, Pre-B, and B cells were lower in mouse–rat chimeras compared to the control (*Figure 2A, B*). In contrast, percentages of CMP-derived erythroid, myeloid and neutrophil progenitors, dendritic cells, and basophils/eosin-ophils were higher (*Figure 2B*). Monocytes and neutrophils were similar, whereas megakaryocytes were decreased in the BM of mouse–rat chimeras (*Figure 2B*). The percentage of LMPPs in mouse–rat chimeras was increased compared to the control (*Figure 2A, B*). HSCs, identified by coexpression of *Kit*, *Ly6a(Sca1)*, and *Flt3* mRNAs (*Rowe et al., 2016*; *Vo and Daley, 2015*), clustered together with myeloid and erythroid progenitors (*Figure 2—figure supplement 4A, B*). The number of ESC-derived HSCs was higher in BM of mouse–rat chimeras compared to the control (*Figure 2—figure supple-ment 4C*), findings consistent with FACS analysis (*Figure 1H, I*). Only 6 out of 6018 BM cells (0.1%) in mouse–rat chimeras contained both mouse and rat mRNA transcripts (*Supplementary files 1 and 2*), indicating that the fusion of mouse and rat BM cells is rare. Thus, although the cellular composition of ESC-derived hematopoietic BM cells was similar in mouse–rat and mouse–mouse chimeras, mouse–rat BM was enriched in HSCs, LMPPs, and CMP-derived erythroid, myeloid, and neutrophil progenitors.

## Single-cell RNA sequencing identifies close similarities in gene expression signatures between ESC-derived hematopoietic cells in mouse–rat and mouse–mouse chimeras

Comparison of gene expression signatures between mouse–rat and mouse–mouse chimeras revealed significant similarities among ESC-derived hematopoietic cell types. Lymphoid progenitors and pro-B cells isolated from mouse–rat and control chimeras expressed *Mif*, *Rcsd1*, and *Tspan13*, whereas pre-B cells expressed *Hmgb2* and *Pgls* (*Figure 2—figure supplement 5A*). *Cd79a* and *CD79b* transcripts were detected in B cells of mouse–rat and control chimeras, whereas *Cd3g* and *Lck* were restricted to T cells (*Figure 2—figure supplement 5A*). Based on the correlation analysis, gene expression profiles of all lymphoid cell types were similar between mouse–rat and control chimeras (*Figure 2—figure supplement 5B*). Likewise, gene expression signatures of myeloid, erythroid, and neutrophil progen-itors and their derivatives in the BM were similar in both experimental groups (*Figure 2—figure supplement 6A, B*). Furthermore, single-cell RNAseq identified close similarities in gene expression signatures of ESC-derived HSCs and LMPPs in both chimeras (*Figure 2—figure supplement 7A, B*).

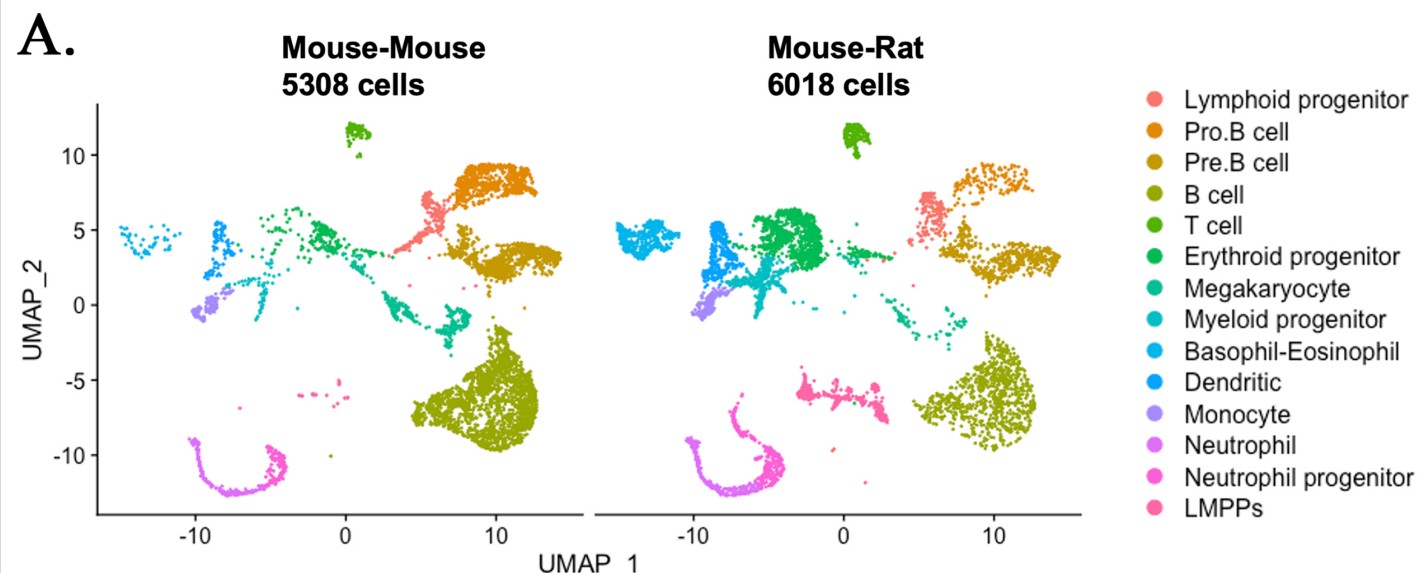

| Cluster | Mouse-Mouse | Mouse-Rat |
|---------|-------------|-----------|
| Lymphoid progenitor | 4.5% | 3.31% |
| Pro.B cell | 14.09% | 2.59% |
| Pre.B cell | 17.29% | 8.99% |
| B cell | 37.09% | 13.43% |
| T cell | 1.75% | 4.04% |
| Erythroid progenitor | 3.84% | 18.64% |
| Myeloid progenitor | 1.34% | 7.73% |
| Neutrophil progenitor | 2.51% | 5.95% |
| Basophil-Eosinophil | 1.02% | 9.02% |
| Dendritic | 1.83% | 6.96% |
| Monocyte | 2.52% | 2.74% |
| Neutrophil | 4.54% | 4.99% |
| Megakaryocyte | 7.2% | 1.68% |
| LMPPs | 0.047% | 0.994% |

**Figure 2.** Single-cell RNAseq analysis identifies embryonic stem cell (ESC)-derived hematopoietic cell lineages in the bone marrow (BM) of mouse–rat chimeras. (**A**) Parallel dimension UMAP plots show identical hematopoietic cell clusters in the BM of mouse–mouse chimera (5308 cells) and mouse–rat chimera (6018 cells). ESC-derived BM cells were obtained from the BM of P10 chimeras using fluorescence-activated cell sorting (FACS) for GFP+ cells, see *Figure 2—figure supplement 1A, B*. Cells from *n* = 3 animals per group were pooled together prior to FACS sorting. Cell clusters were identified from single-cell RNAseq datasets using Uniform Manifold Approximation and Projection (UMAP) method, see also *Figure 2—figure supplements 2A, B and 3*. Hematopoietic stem cells (HSCs) were identified by coexpression of *Kit*, *Ly6a* (*Sca1*), and *Flt3* (*Flk2*), see *Figure 2—figure supplement 4*. Heatmaps and linear regression analysis identified significant similarities in gene expression signatures of lymphoid and myeloid progenitor cells obtained from mouse–rat (R) and mouse–mouse chimeras (M), see *Figure 2—figure supplement 5A, B* and *Figure 2—figure supplement 6A, B*.

*Figure 2 continued on next page*

*Figure 2 continued*

Gene expression profiles of ESC-derived HSCs and lymphoid-primed multipotent progenitor cells are shown in *Figure 2—figure supplement 7A, B*. (**B**) Table shows percentages of cells in individual clusters in mouse–mouse and mouse–rat chimeras. Blue color indicates decreased percentages of cells in mouse–rat chimeras compared to mouse–mouse chimeras. Red color indicates increased percentages of cells in mouse–rat chimeras.

The online version of this article includes the following figure supplement(s) for figure 2:

**Figure supplement 1.** Purification of mouse embryonic stem cell (ESC)-derived cells from bone marrow (BM) of mouse–rat and mouse–mouse chimeras before scRNAseq.

**Figure supplement 2.** Single-cell RNAseq analysis identifies hematopoietic cell subsets in the bone marrow (BM) of mouse–rat chimeras.

**Figure supplement 3.** Violin plots confirm expression of hematopoietic marker genes in bone marrow (BM) cell clusters.

**Figure supplement 4.** Single-cell RNAseq analysis identifies genes expressed in hematopoietic stem cells (HSCs) in chimeric bone marrow (BM).

**Figure supplement 5.** Embryonic stem cell (ESC)-derived lymphoid cell types in mouse–rat and mouse–mouse chimeras exhibit identical gene expression profiles.

**Figure supplement 6.** Embryonic stem cell (ESC)-derived myeloid cell types in mouse–rat and mouse–mouse chimeras exhibit similar gene expression profiles.

**Figure supplement 7.** Heatmaps identify gene expression profile of embryonic stem cell (ESC)-derived hematopoietic stem cells (HSCs) and lymphoid-primed multipotent progenitor (LMPP) cells from mouse–rat and mouse–mouse chimeras.

Thus, gene expression signatures of ESC-derived hematopoietic cells were similar in mouse–rat and control mouse–mouse chimeras.

## Chimeric BM is enriched in mouse hematopoietic progenitor cells and rat chondrocytes

To examine the composition and origin of stromal cells in mouse–rat chimeras, we used an enzymatic digestion to obtain both hematopoietic and stromal cells from BM of P5 mouse–rat chimeras and compared them to BM cells of mice and rats of the same age. Flow sorting for GFP was performed to separate donor mouse cells (GFP⁺) and recipient rat cells (GFP⁻) in the chimeric BM. BM from control P5 mice and rats was also FACS-sorted for GFP⁻ BM cells to ensure similar conditions of cell preparations prior to single-cell RNAseq. Based on published gene expression signatures (*Baccin et al., 2020*), 6375 mouse and 5495 rat cells were identified in the chimeras, which were compared to 6418 cells from control mice and 7016 cells from control rats. Similar hematopoietic and stromal cell clusters were present in BM of mice, rats, and mouse–rat chimeras (*Figure 3A–C*). These included stromal cell clusters (endothelial cells, fibroblasts, myofibroblasts, and chondrocytes) and hematopoietic cell clusters with various progenitor and differentiated hematopoietic cell types. Since we did not enrich BM cell populations for Lin⁻ cells, some rare BM cell subsets, such as HSCs, LMPPs, and dendritic cells, were not detected as separate cell clusters. Compared to normal BM from P5 mice, chimeric BM was enriched in mouse ESC-derived hematopoietic progenitor cells, such as myeloid, granulocyte, and erythroid progenitors, whereas mouse-derived B cell lineages were reduced (*Figure 3A*), findings consistent with single-cell RNAseq comparison of P10 BM from mouse–rat and mouse–mouse chimeras (*Figure 2*). The percentage of mouse endothelial cells was increased in mouse–rat BM, whereas the percentages of mouse chondrocytes and fibroblasts were reduced compared to mouse control (*Figure 3A*). In contrast, mouse–rat BM was enriched in rat-derived chondrocytes and fibroblasts, but the percentages of endothelial and most hematopoietic cells were reduced compared to age-matched rats (*Figure 3B*). Thus, mouse cells preferentially contributed to hematopoietic progenitors and endothelial cells, whereas rat cells contributed to the majority of chondrocytes and fibroblasts.

Direct comparison of mouse and rat cells within chimeric BM demonstrated significant similarities between gene expression signatures of hematopoietic and stromal cell lineages (*Figure 3—figure supplement 1A–D*). To examine cell signaling between hematopoietic progenitors and stromal cells in BM of mouse–rat chimeras, we generated the map of potential ligand–receptor interactions using P5 single-cell RNAseq datasets. There were remarkable similarities in major receptor–ligand interactions between stromal and erythro-myeloid progenitor cells (EMPs) (*Figure 4*). Regardless of mouse and rat origins of BM cells, endothelial cells interacted with EMPs through the *Cxcl12-Cxcr4* receptor–ligand signaling pair. The main signaling circuit between fibroblasts and EMPs was *Lama2-Itga6*, whereas chondrocytes signaled to EMPs through *App-Itga6* and *Comp-Cd47* pathways (*Figure 4*). Major receptor–ligand interactions between granulocyte–monocyte progenitor (GMP) cells and stromal

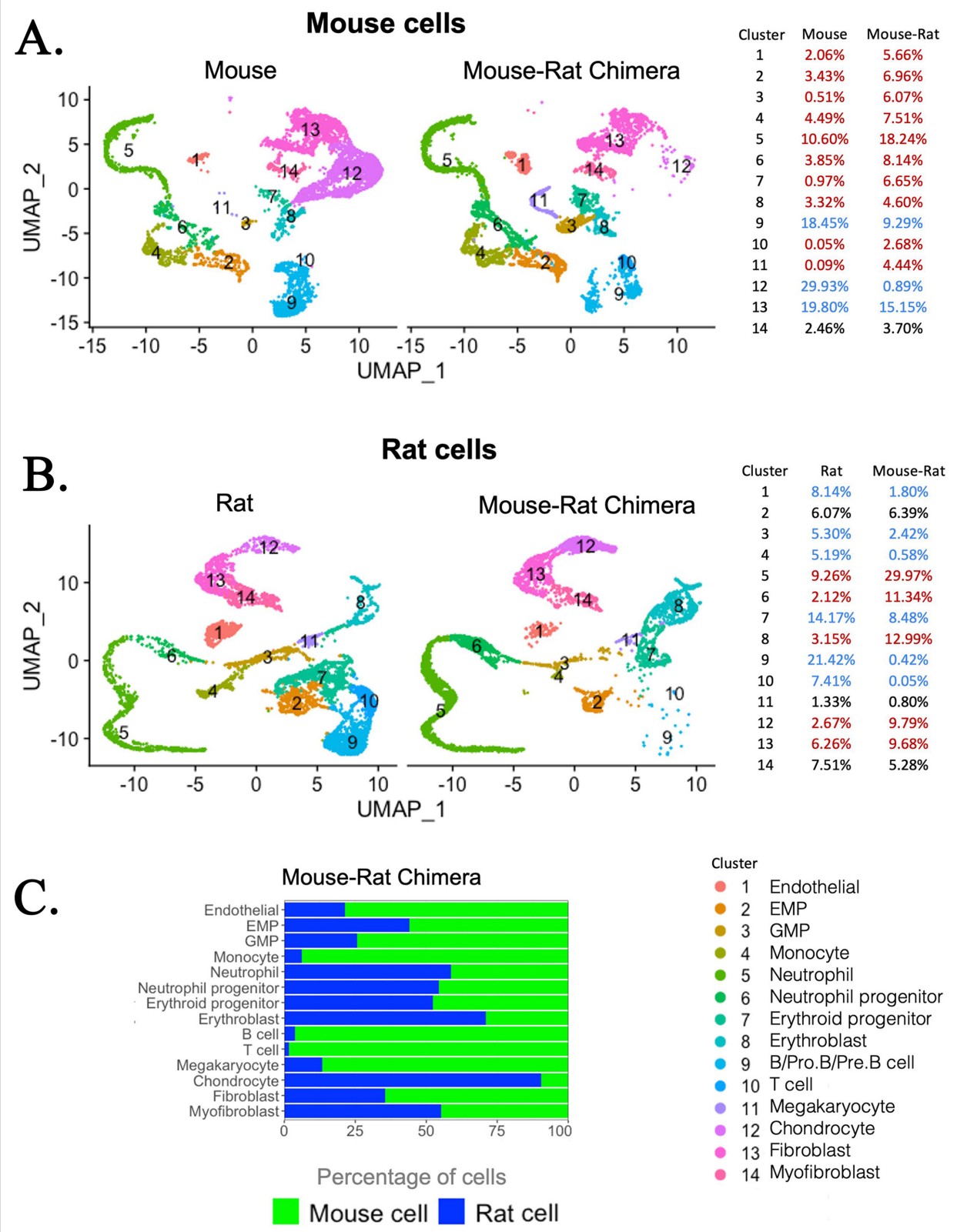

**Figure 3.** Single-cell RNAseq analysis shows increased percentages of embryonic stem cell (ESC)-derived hematopoietic progenitors and endothelial cells but decreased percentages of ESC-derived chondrocytes in the bone marrow (BM) of mouse–rat chimeras. (**A, B**) Parallel dimension UMAP plots show identical hematopoietic and stromal cell clusters in the BM of P5 mice, rats, and mouse–rat chimeras. BM cells were obtained from P5 animals using an enzymatic digestion (*n* = 5 animals per group) and pooled prior to single-cell RNAseq. Cell clusters were identified from single-cell RNAseq

*Figure 3 continued on next page*

*Figure 3 continued*

datasets using Uniform Manifold Approximation and Projection (UMAP) method. Red color in the tables indicates increased percentages of cells in mouse–rat chimeras compared to either mice or rats of the same age. Blue color indicates decreased percentages of cells in mouse–rat chimeras. Gene expression signatures of mouse and rat hematopoietic and stromal cells are shown in *Figure 3—figure supplement 1A–D*. (**C**) A bar graph shows relative percentages of ESC-derived mouse cells (green) and endogenous rat cells (blue) in the BM of P5 mouse–rat chimeras.

The online version of this article includes the following figure supplement(s) for figure 3:

**Figure supplement 1.** Heatmaps compare gene expression profile in mouse and rat hematopoietic and stromal cells that form bone marrow (BM) in mouse–rat chimeras.

cells were also similar in BM cells of mouse and rat origin (*Figure 4—figure supplement 1*). These include *Cxcl12-Cxcr4* signaling between endothelial cells and GMPs, *Col1a1-Cd44* signaling between fibroblasts and GMPs, and *App-Il18rap* signaling between chondrocytes and GMPs (*Figure 4—figure supplement 1*). Analysis of expression patterns for several ligands and their receptors revealed no obvious differences between mouse and rat cells (*Figure 4—figure supplement 2*). These results demonstrate that mouse and rat BM cells use similar signaling pathways between stromal and hematopoietic progenitor cells.

## Mouse HSCs in mouse–rat chimeras develop earlier than rat HSCs

Fetal HSCs emerge from hemogenic endothelium in the aorta–gonad–mesonephros region and later undergo expansion in the embryonic liver (*Gao et al., 2018*; *Weijts et al., 2021*). To examine the development of HSCs in mouse–rat chimeras, mouse-derived (GFP+) and rat-derived (GFP−) hemogenic endothelial cells were visualized in the dorsal aorta by colocalization of FLK1 with RUNX1 transcription factor (*Figure 5A, B*). At E11, mouse embryos were significantly larger than rat and mouse–rat chimeric embryos (*Figure 5—figure supplement 1*), consistent with previous studies demonstrating that the main stages of mouse embryonic development occur approximately 1.5 days faster compared to embryonic development in the rat (*Farrington-Rock et al., 2008*; *Marcela et al., 2012*; *Takahashi and Osumi, 2005*; *Torres et al., 2008*). Therefore, we compared E11 mouse embryos with E12.5 rat and chimeric embryos which were in similar developmental stages. In the dorsal aorta of mouse–rat chimeras, the majority of FLK1+RUNX1+ cells expressed GFP, indicating the mouse origin of these cells (*Figure 5B*). Later in development, percentages of mouse Lin− cells, LSKs, and ST-HSCs were higher in fetal livers of mouse–rat chimeras as demonstrated by FACS analysis for Lin, CD117, Sca1, CD48, and CD150 (*Figure 5C* and *Figure 5—figure supplement 2*). The percentage of LT-HSC in fetal livers was unchanged (*Figure 5C*). Thus, ESC complementation causes the earlier development of donor HSCs in the dorsal aorta and increases percentages of donor-derived Lin− cells, LSKs, and ST-HSCs in the fetal liver.

## Transplantation of ESC-derived BM cells from interspecies mouse–rat chimeras rescues lethally irradiated syngeneic mice

To test functional properties of mouse BM hematopoietic progenitor cells derived through a rat, cells were FACS-sorted for GFP from the BM of juvenile mouse–rat chimeras and transferred into the tail vein of syngeneic C57BL/6 adult mice that received the lethal dose of whole-body gamma-irradiation 3 hr prior to the BM transplant (*Figure 6A*). Consistent with published studies (*Rowe et al., 2016*; *Sugimura et al., 2017*; *Vo and Daley, 2015*), all mice without BM transplant died between 9 and 12 days after irradiation (*Figure 6B*). In contrast, all 20 mice transplanted with GFP+ BM cells from mouse–rat chimeras survived after lethal irradiation (*Figure 6B, C*). Histological assessment of femur bones confirmed the presence of GFP+ donor cells in the BM compartment of transplanted mice (*Figure 6D*). Blood analysis of mice harvested 8 days after irradiation showed significant decreases in white blood cells (WBCs), red blood cells (RBCs), platelets (PLT), hemoglobin (Hb) as well as numbers of granulocytes, monocytes, and lymphocytes (*Figure 7A* and *Figure 7—figure supplements 1 and 2*). Transplantation of ESC-derived BM cells from mouse–rat chimeras increased WBC and the numbers of granulocytes, monocytes, and lymphocytes in the peripheral blood at day 8 (*Figure 7A* and *Figure 7—figure supplements 1 and 2*). Contribution of ESC-derived BM cells to granulocytes, monocytes, and B cells was higher compared to erythroid and T cells (*Figure 7B* and *Figure 7—figure supplement 3*). At 5 months after BM transplantation, ESC-derived cells completely restored blood

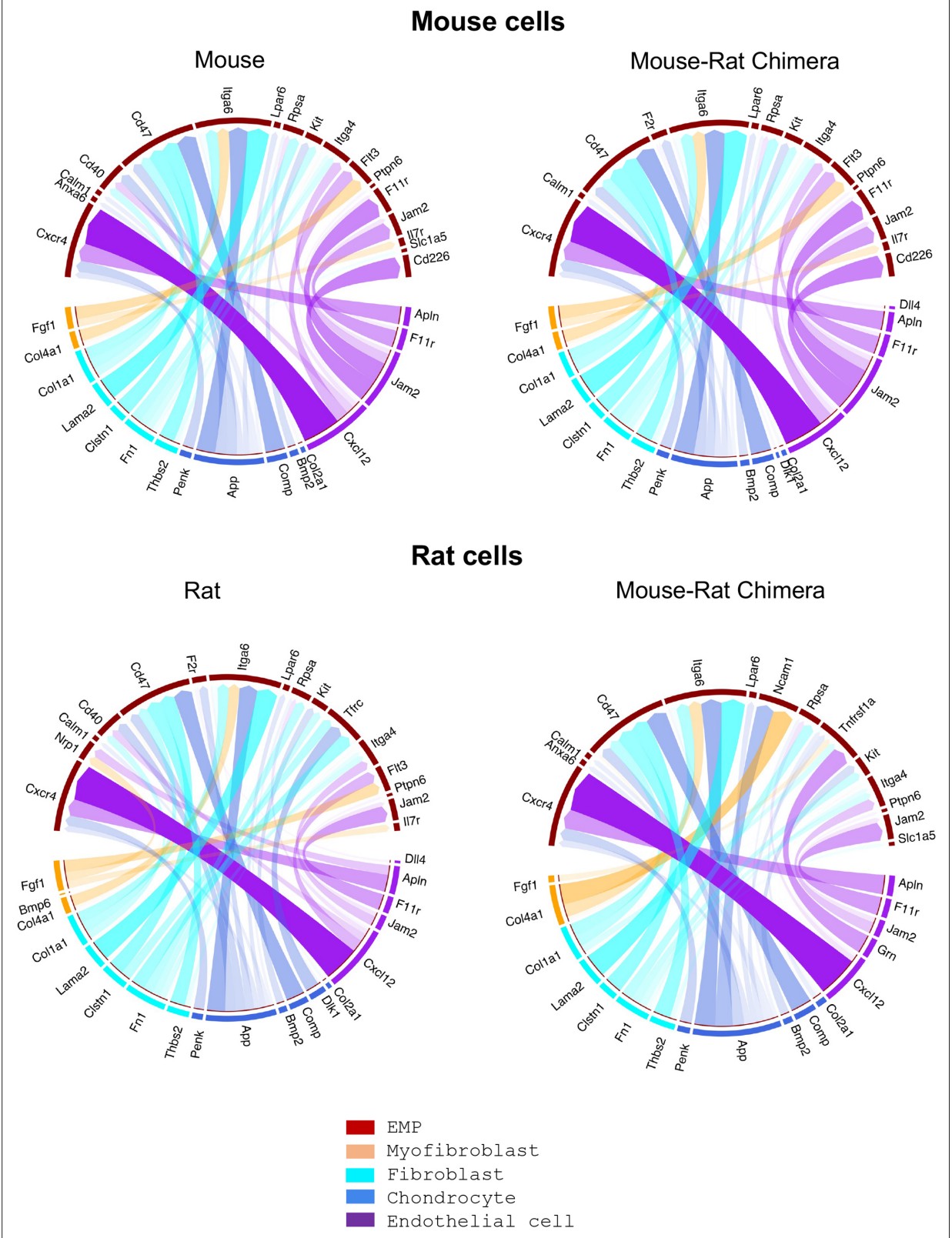

**Figure 4.** Single-cell RNAseq analysis shows remarkable similarities in major receptor–ligand interactions between erythro-myeloid progenitors and stromal cells of mouse and rat origins. Bone marrow (BM) cells were obtained from P5 animals using an enzymatic digestion (*n* = 5 animals per group). Single-cell RNAseq was performed to identify BM stromal and erythro-myeloid progenitor cells (EMPs) based on gene expression signatures. The R package *NicheNet* was used to analyze the expression of ligands and receptors to identify intercellular communication patterns between EMPs and

*Figure 4 continued on next page*

*Figure 4 continued*

BM stromal cells. Receptor–ligand interactions between stromal and granulocyte–monocyte progenitor (GMP) cells are shown in *Figure 4—figure supplement 1*. Violin plots were used to identify expression of ligands and their receptors in hematopoietic and stromal BM cells, see *Figure 4—figure supplement 2*.

The online version of this article includes the following figure supplement(s) for figure 4:

**Figure supplement 1.** Single-cell RNAseq analysis shows similar receptor–ligand interactions between stromal and granulocyte–monocyte progenitor (GMP) cells.

**Figure supplement 2.** Violin plots show expression of ligands and their receptors in hematopoietic and stromal bone marrow (BM) cells.

cell numbers, PLT and Hb in lethally irradiated mice (*Figure 7C* and *Figure 7—figure supplements 1 and 2*). Long-term contributions of ESC-derived BM cells to all hematopoietic cell lineages in the peripheral blood were between 49% and 96% (*Figure 7C* and *Figure 7—figure supplement 3*). Thus, transplantation of ESC-derived BM cells from mouse–rat chimeras prevented mortality and restored hematopoietic blood lineages in lethally irradiated syngeneic mice.

## Transplantation of ESC-derived BM cells from interspecies mouse–rat chimeras resulted in the long-term contribution of donor cells to hematopoietic progenitor cells

Based on FACS analysis of irradiated mice at day 8, whole-body irradiation decreased the number of hematopoietic progenitor cells in the BM, including LSKs, ST-HSCs, and LT-HSCs (*Figure 7D* and *Figure 7—figure supplement 4A, B*). Transplantation of ESC-derived BM cells significantly increased LSKs but did not affect the numbers of ST-HSCs and LT-HSCs in irradiated mice (*Figure 7D*). Contribution of ESC-derived BM cells to Lin⁻ and LSK cell subsets was high, whereas ESC contribution to ST-HSCs and LT-HSCs at day 8 was low (*Figure 7E* and *Figure 7—figure supplement 5*). At 5 months after BM transplantation, percentages of LSKs, ST-HSCs, and LT-HSCs in the BM were increased (*Figure 7D* and *Figure 7—figure supplement 4B*). Long-term contribution of ESC-derived BM cells to LSKs, ST-HSCs, and LT-HSCs was between 92% and 95% (*Figure 7F* and *Figure 7—figure supplement 5*). Finally, we performed BM transplantation again in secondary recipients to establish the functional potential and self-renewal capacity of the chimeric HSCs (*Figure 7—figure supplement 6A*). The secondary BM transplantation rescued lethally irradiated mice and resulted in long-term engraftment of ESC-derived HSCs into hematopoietic cell lineages in the BM and peripheral blood (*Figure 7—figure supplement 6B–E*). Altogether, transplantation of ESC-derived BM cells from mouse–rat chimeras resulted in efficient, long-term contribution of donor cells to the BM and blood of lethally irradiated mice.

## Discussion

Recent single-cell RNA sequencing studies identified remarkable diversity of hematopoietic cell types in the BM (*Baccin et al., 2020*). Generation of functional BM cells from pluripotent ESCs or iPSCs in a dish or in organoids represents a formidable challenge (*Rowe et al., 2016*; *Vo and Daley, 2015*). In the present study, we used blastocyst complementation to generate a diversity of hematopoietic cell types from mouse ESCs in rat embryos. Interspecies mouse–rat chimeras were viable and contained approximately 25% of ESC-derived mouse cells in the BM. It is possible that inactivation of genes critical for hematopoiesis in rat embryos prior to blastocyst complementation can improve the integration of mouse ESCs into the BM of mouse–rat chimeras. This approach was supported by recent studies with mouse–mouse chimeras, in which ESCs contributed to more than 90% of hematopoietic cells in mice deficient for either Kit or Flk1 (*Hamanaka et al., 2018*; *Jansson and Larsson, 2010*). While ESCs contributed to all hematopoietic cell lineages in interspecies BM, the percentage of lymphoid progenitors was lower, whereas the percentages of myeloid progenitor cells and HSCs were higher in mouse–rat chimeras compared to control mouse–mouse chimeras. Since both chimeras were produced by complementing blastocysts with mouse ESCs from the same ESC-GFP cell line, it is unlikely that these changes are dependent on donor ESCs. It is possible that the observed differences in BM cellular composition between mouse–rat and mouse–mouse chimeras are due to interactions of donor ESCs with the host embryo. Structural and functional differences between hormones, growth

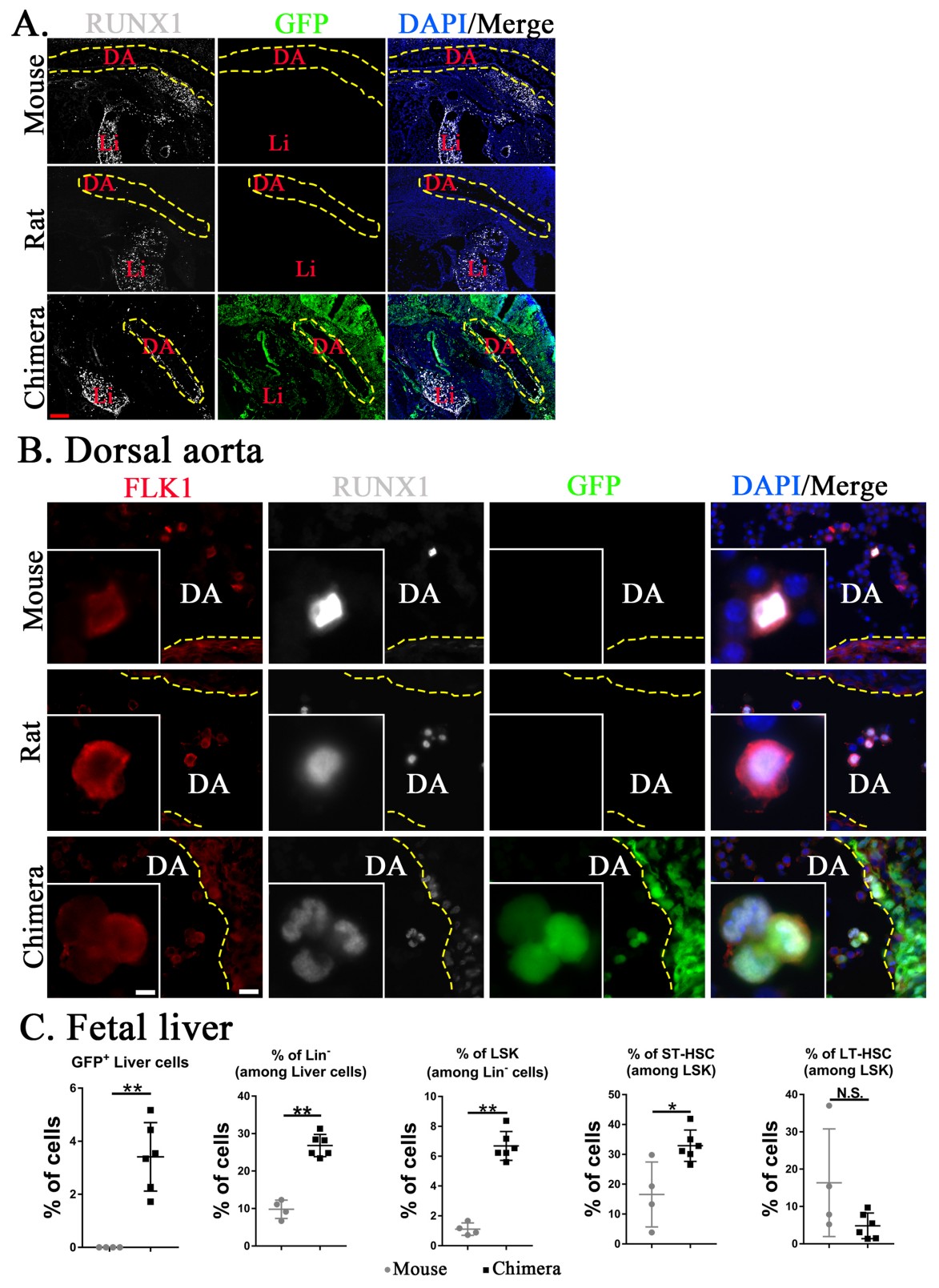

**Figure 5.** Mouse hematopoietic stem cells (HSCs) in mouse–rat chimeras develop earlier than rat HSCs. (**A, B**) Immunostaining for RUNX1 (white) and FLK1 (red) shows that hemogenic endothelium in the dorsal aorta (DA) of mouse–rat chimeras develops mostly from embryonic stem cell (ESC)-derived mouse cells. GFP (green) was used to identify ESC-derived cells, whereas 4′,6-diamidino-2-phenylindole (DAPI, blue) was used to stain cell nuclei. Frozen sections were obtained from E11 mouse embryos and E12.5 embryos from rats and mouse–rat chimeras since these embryos are in similar

*Figure 5 continued on next page*

*Figure 5 continued*

developmental stages, see also *Figure 5—figure supplement 1A–C*. DA indicates the lumen of dorsal aorta. Yellow dashed line indicates the luminal surface of DA wall. Inserts show high magnification of hemogenic endothelial cells expressing both RUNX1 and FLK1. Scale bars are: A, 200 μm; B, 20 μm; inserts in B, 5 μm. Abbreviations: DA, dorsal aorta; Li, liver. (**C**) Fluorescence-activated cell sorting (FACS) analysis shows increased percentages of mouse ESC-derived Lin⁻ cells, LSKs, and short-term HSCs (ST-HSCs) in fetal livers of mouse–rat chimeras (*n* = 6) compared to control mouse embryos (*n* = 4), see also *Figure 5—figure supplement 2*. Fetal livers were obtained from E15.5 mouse–rat chimeras and E14 mouse embryos since these embryos are in similar developmental stages. *p < 0.05, **p < 0.01, N.S. indicates no significance, see also *Source data 1*.

The online version of this article includes the following figure supplement(s) for figure 5:

**Figure supplement 1.** Mouse embryonic development occurs faster than embryonic development in the rat and mouse–rat chimera.

**Figure supplement 2.** Fluorescence-activated cell sorting (FACS) analysis identifies mouse hematopoietic progenitor cells in fetal livers of mouse–rat chimeras.

factors, and their receptors in rats and mice can contribute to the efficiency or timing of differentiation of mouse ESCs into hematopoietic cell lineages in BM of chimeras.

Our data demonstrate that chimeric HSCs develop more efficiently from donor mouse cells in the dorsal aorta, fetal liver, and BM, whereas rat cells are less efficient to differentiate into HSCs. Since we observed high numbers of mouse hemogenic endothelial cells in the chimeric dorsal aorta, it is likely that donor hemogenic endothelium undergoes direct transition to functional HSCs in the fetal liver, whereas endogenous (non-GFP+) hemogenic endothelium can be a source of rat HSCs. Since mouse embryos develop faster compared to rat embryos by approximately 1.5 days (*Farrington-Rock et al., 2008*; *Marcela et al., 2012*; *Takahashi and Osumi, 2005*; *Torres et al., 2008*), it is possible that mouse ESC-derived progenitor cells migrate faster into developing hematopoietic niches in the mouse–rat chimeras, leading to preferential development of HSCs from cells of mouse origin and contributing to increased numbers of mouse-derived hematopoietic progenitors in the BM of mouse–rat chimeras. These data suggest that using donor ESCs from species with less gestational time in interspecies 'bioreactors' can lead to larger quantities of ESC-derived hematopoietic progenitors in the chimeric BM. Our single-cell RNAseq analysis enabled us to identify potential signaling pathways and receptor–ligand interactions between hematopoietic progenitors and stromal cells in the BM. These pathways include Cxcl12-Cxcr4 signaling between hematopoietic progenitors and endothelial cells, which plays a critical role in maintenance of HSCs during BM homeostasis and promotes niche regeneration and hematopoietic reconstitution after BM transplantation (*Baccin et al., 2020*; *Singh et al., 2020*; *Sugiyama et al., 2006*). Other pathways identified in our studies, including Lama2-Itga6, App-Itga6, Comp-Cd47, Col1a1-Cd44, and App-Il18rap, have not been extensively studied in the BM microenvironment but are implicated in regulation of cell adhesion, migration, oncogenesis, fibrosis, and inflammatory responses (*Kiratipaiboon et al., 2020*; *Sibin et al., 2019*; *Rock et al., 2010*; *Strelnikov et al., 2021*; *Yang et al., 2017*). Notably, our data suggest that some of these signaling pathways can be targeted to modulate the development and expansion of donor ESC-derived hematopoietic progenitor cells in the BM of interspecies chimeras.

Despite mosaicism in interspecies BM, mouse ESC-derived cells from multiple hematopoietic cell lineages were highly differentiated and indistinguishable from the normal mouse BM cells based on gene expression signatures and cell surface proteins. Consistent with functional competency of ESC-derived BM, transplantation of BM cells into lethally irradiated syngeneic mice prevented mortality and resulted in long-term contribution of ESC-derived cells to all hematopoietic cell lineages in the BM and peripheral blood. One of the limitations of our studies is that the functional potential of chimeric HSCs was established from whole BM transplants and not from transplantation of purified HSCs. While these experiments are technically challenging, transplantation of FACS-sorted donor HSCs into lethally irradiated mice will be needed in our future studies to investigate whether chimeric HSCs are fully functional to restore all hematopoietic cell lineages after irradiation. Our results are consistent with recent studies demonstrating the ability of mouse ESCs to generate functional pancreatic, endothelial, and kidney cells in interspecies mouse–rat chimeras (*Goto et al., 2019*; *Wang et al., 2021*; *Yamaguchi et al., 2017*). Interestingly, long-term contribution of donor BM cells to ST-HSCs and LT-HSCs of irradiated mice was high, supporting the ability of donor HSCs to self-renew. In contrast, the short-term contribution of donor BM cells to ST-HSCs and LT-HSCs of irradiated mice was low. Low contribution of donor BM to HSCs at day 8 is not surprising considering an acute hematopoietic deficiency in lethally irradiated mice. It is possible that most donor-derived HSCs undergo rapid

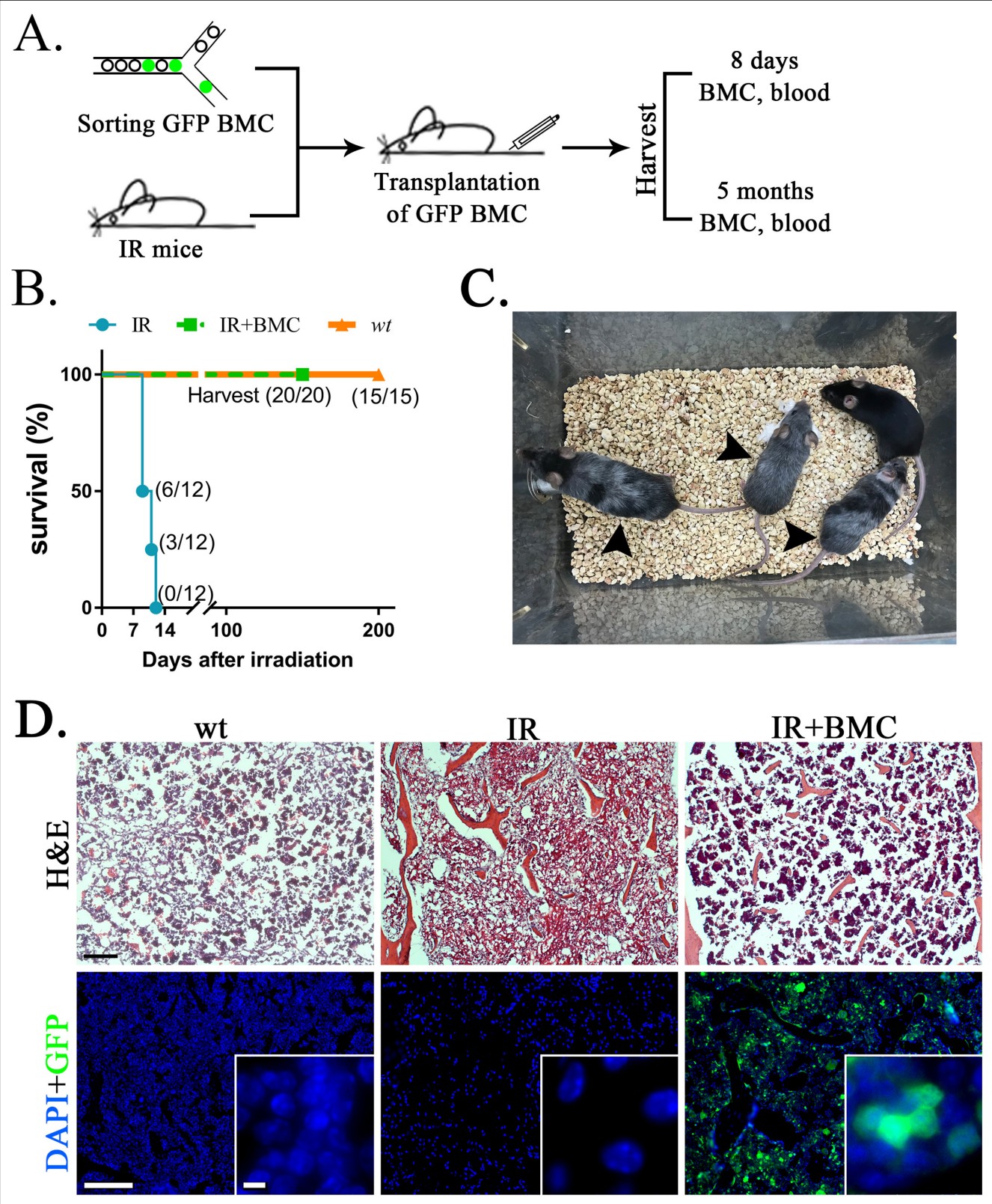

**Figure 6.** Transplantation of mouse embryonic stem cell (ESC)-derived bone marrow (BM) cells from interspecies mouse–rat chimeras rescues lethally irradiated syngeneic mice. (**A**) Schematic diagram shows transplantation of ESC-derived bone marrow cells (BMCs) into lethally irradiated (IR) mice. ESC-derived cells were obtained from the BM of juvenile mouse–rat chimeras using fluorescence-activated cell sorting (FACS) for GFP⁺ cells. BM and peripheral blood were harvested 8 days and 5 months after BM transplantation. (**B**) Kaplan–Meier survival analysis shows a 100% mortality in irradiated

*Figure 6 continued on next page*

*Figure 6 continued*

mice. Survival is dramatically improved after transplantation of irradiated mice with ESC-derived BM cells obtained from mouse–rat chimeras (IR + BMC). Survival in untreated wild-type (wt) mice is shown as a control (*n* = 12–20 mice in each group). (**C**) Photograph shows irradiated C57BL/6 mice 5 months after successful BM transplantation. Untreated C57BL/6 mouse is shown as a control. Gray color of irradiated mice (arrows) is consistent with large doses of whole-body radiation treatment. (**D**) Hematoxylin and eosin (H&E) staining shows increased amounts of hematopoietic cells in femur bones after BM transplantation into irradiated mice (top panels). GFP⁺ donor cells (green) are abundant in the BM compartment of transplanted mice (bottom panels). DAPI (blue) was used for counterstaining. Scale bars are: D, 200 µm; inserts in D, 5 µm.

differentiation into other hematopoietic cell types to compensate for the loss of injured hematopoietic cells after irradiation.

Generation of intraspecies chimeras through blastocyst complementation creates an interesting opportunity to use patient-derived iPSCs to produce tissues or even organs in large animals, for example, pigs or sheep, which can serve as 'biological reactors'. However, at this stage of technological advances it is impossible to restrict the integration of ESC/iPSC-derived cells into selected organs or cell types. Off-target integration of ESCs and iPSCs into the brain, testes, and sensory organs raises important ethical concerns for the use of human–animal chimeras in regenerative medicine (*Masaki and Nakauchi, 2017*; *Wu et al., 2016*). To improve the selectivity of ESC/iPSC integration into chimeric tissues, various genetic modifications can be introduced into the host embryos to advance the technology. Harvest of tissues from chimeric embryos instead of adult chimeras can alleviate some of the ethical concerns, suggesting a possibility of using chimeric embryos as a potential source of patient-specific hematopoietic progenitor cells.

In summary, blastocyst complementation of rat embryos with mouse ESCs was used to simultaneously generate multiple hematopoietic and stromal cell lineages in the BM. ESC-derived cells in mouse–rat chimeras were indistinguishable from normal mouse BM cells based on gene expression signatures and cell surface markers. Transplantation of ESC-derived BM cells rescued lethally irradiated syngeneic mice and resulted in long-term contribution of donor cells to hematopoietic cell lineages. Thus, the interspecies chimeras could be considered for in vivo differentiation of patient-derived iPSCs into hematopoietic cell lineages for future cell therapies.

## Materials and methods
### Mice, rats, and generation of mouse–rat and mouse–mouse chimeras through blastocyst complementation

C57BL/6 mice were purchased from Jackson Lab. Interspecies mouse–rat chimeras were generated using blastocyst complementation as described (*Li et al., 2021*; *Wang et al., 2021*). Briefly, blastocysts from SD rats were obtained at embryonic day 4.5 (E4.5), injected with 15 GFP-labeled mouse ESC cells (ESC-GFP, C57BL/6 background) (*Sun et al., 2021*; *Wen et al., 2021*) and transferred into pseudopregnant SD rat females. Mouse–mouse chimeras were generated by complementing CD1 blastocysts with 15 mouse ESC-GFP cells. For FACS analysis and BM transplantation, BM cells were collected from chimeric pups that were harvested between postnatal day 4 (P4) and P10. For single-cell RNA sequencing, BM cells were prepared from P10 and P5 mice, rats, and chimeras. To perform BM transplantation, BM cells from two tibias and two fibulas of mouse–rat chimeras were collected and FACS-sorted for ESC-derived (GFP⁺) cells. 500,000 FACS-sorted GFP⁺ BM cells were intravenously (i.v.) injected into lethally irradiated C57BL/6 male mice (6–8 weeks of age) via the tail vein. Three hours before BM transplantation, whole-body irradiation was performed using 11.75 Gy. Mice were harvested after 8 days or 5 months after BM transplantation. For the second BM transplantation, GFP⁺ BM cells were FACS-sorted from irradiated mice 5 months after the first BM transplantation and then i.v. injected into new irradiated recipients. Tissue dissection, processing, and preparation of single-cell suspensions were carried out as described (*Bolte et al., 2011*; *Kalin et al., 2008*; *Kalinichenko et al., 2003*; *Kim et al., 2005*; *Wang et al., 2003*). Blood analysis was performed in animal facility of Cincinnati Children's Hospital Research Foundation.

### Single-cell RNAseq analysis of ESC-derived BM cells

Prior to scRNAseq (10× Chromium platform), BM cells were pooled from three P10 mouse–rat chimeras and three P10 mouse–mouse (control) chimeras and then FACS-sorted for GFP and the *lineage* (Lin)

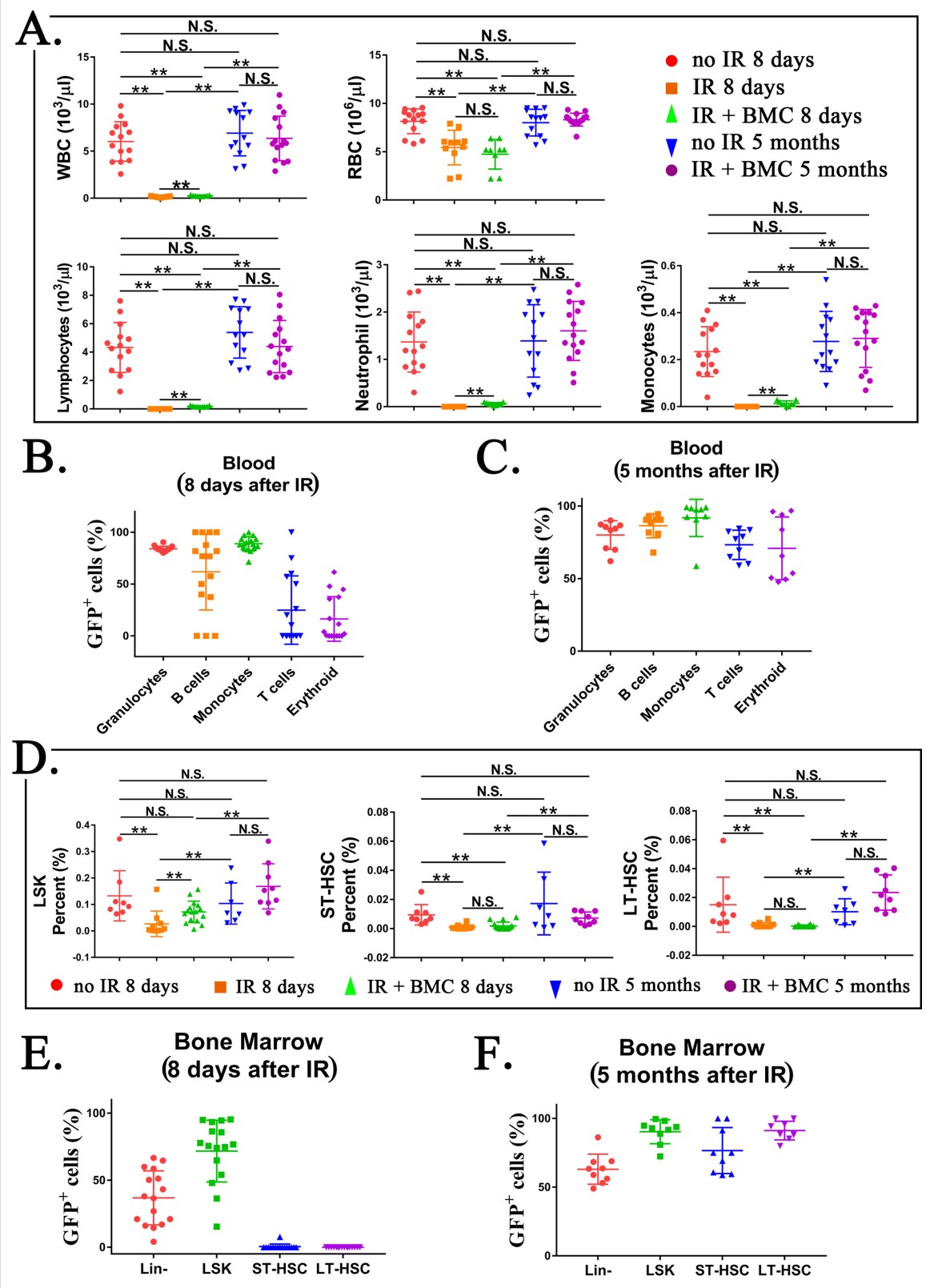

**Figure 7.** Transplantation of mouse embryonic stem cell (ESC)-derived bone marrow (BM) cells from interspecies mouse–rat chimeras restores hematopoietic cell lineages in the blood and BM of lethally irradiated syngeneic mice. (**A**) Blood analysis shows that transplantation with ESC-derived BM cells from mouse–rat chimeras increases white blood cell (WBC) counts and red blood cell (RBC) counts in the peripheral blood of irradiated recipients. Blood samples were obtained from untreated mice (no IR), lethally irradiated mice without BM transplant (IR), and lethally irradiated mice with

*Figure 7 continued on next page*

*Figure 7 continued*

BM transplant (IR + BMC). BM transplantation was performed using ESC-derived BM cells obtained from juvenile mouse–rat chimeras. Fluorescence-activated cell sorting (FACS) analysis of the peripheral blood to identify granulocytes, B cells, monocytes, T cells, and erythroid cells in shown in *Figure 7—figure supplement 1*. Concentrations of lymphocytes, monocytes, and neutrophil in the blood were increased after BM transplantation (*n* = 9–15 mice in each group), **p < 0.01, N.S. indicates no significance, see also *Source data 1*. BM transplantation also increased concentrations of platelets, hemoglobin, basophils, and eosinophils in the peripheral blood, see *Figure 7—figure supplement 2*. (B, C) FACS analysis for GFP$^+$ cells in each cell subset shows that ESC-derived BM cells from mouse–rat chimeras contribute to multiple hematopoietic cell lineages in the peripheral blood of lethally irradiated mice (*n* = 9–16 mice in each group), see also *Figure 7—figure supplement 3*. (D) FASC analysis shows that transplantation with ESC-derived BM cells from mouse–rat chimeras increases percentages of LSKs, short-term HSCs (ST-HSCs), and long-term HSCs (LT-HSCs) in the BM of irradiated mice 5 months after BM transplantation (*n* = 9–16 mice in each group), see also *Figure 7—figure supplement 4A, B*. **p < 0.01, N.S. indicates no significance, see also *Source data 1*. (E, F) FACS analysis for GFP$^+$ shows that ESC-derived BM cells from mouse–rat chimeras contribute to multiple hematopoietic progenitor cells in the BM of irradiated mice (*n* = 9–16 mice in each group), see also *Figure 7—figure supplement 5*. For secondary transplantation of mouse ESC-derived BM cells into lethally irradiated syngeneic mice, see *Figure 7—figure supplement 6A–E*.

The online version of this article includes the following figure supplement(s) for figure 7:

**Figure supplement 1.** Fluorescence-activated cell sorting (FACS) analysis identifies granulocytes, B cells, monocytes, T cells, and erythroid cells in the peripheral blood after bone marrow (BM) transplantation.

**Figure supplement 2.** Transplantation of irradiated mice with embryonic stem cell (ESC)-derived bone marrow (BM) cells from mouse–rat chimeras increases hemoglobin (Hb) concentration and numbers of platelets (PLT), basophils, and eosinophils in the peripheral blood.

**Figure supplement 3.** Identification of embryonic stem cell (ESC)-derived cells in the peripheral blood of irradiated mice after bone marrow (BM) transplantation.

**Figure supplement 4.** Transplantation of mouse embryonic stem cell (ESC)-derived bone marrow (BM) cells from interspecies mouse–rat chimeras results in reconstitution of BM hematopoietic and progenitor cells after irradiation.

**Figure supplement 5.** Identification of embryonic stem cell (ESC)-derived hematopoietic cells in the bone marrow (BM) of irradiated mice after BM transplantation.

**Figure supplement 6.** Secondary transplantation of mouse embryonic stem cell (ESC)-derived bone marrow (BM) cells from interspecies mouse–rat chimeras rescues lethally irradiated syngeneic mice.

marker. Since the numbers of HSCs and other hematopoietic progenitors in BM are significantly low compared to numbers of differentiated hematopoietic cells, the cell mixtures were enriched for BM progenitor cell populations by combining 90% of FACS-sorted GFP$^+$Lin$^-$ cells and 10% of GFP$^+$Lin$^+$ cells in each experimental group. This enrichment enabled us to obtain enough progenitor cells for UMAP clustering analysis. In separate scRNAseq experiments, all BM cells (including hematopoietic, vascular, and stromal cells) were prepared from P5 mice, rats, and mouse–rat chimeras using enzymatic digestion and cell purification as described (*Baccin et al., 2020*). BM cells from five animals were pooled together prior to single-cell RNAseq. All raw data and the processed count matrix of BM datasets were uploaded to the GEO database (accession number GSE184940). Read alignments, quality controls, and false discovery rates were described previously (*Guo et al., 2019*; *Ren et al., 2019*; *Wang et al., 2022*). Identification of cell clusters and quantification of cluster-specific gene expression in BM scRNAseq datasets were performed as described (*Baccin et al., 2020*; *Wang et al., 2021*; *Wen et al., 2021*). To assess the transcriptomic similarity of ESC-derived and endogenous BM cells, the scRNAseq datasets were normalized with *SCTransform* and then integrated utilizing the canonical correlation analysis. In the integrated scRNAseq datasets, the *SelectIntegrationFeatures* in Seurat package (version 4.0.0 in R 4.0 statistical environment) was used to identify anchors for integration. The *RunPCA* function was used for principal component analysis (PCA) of scRNAseq datasets, and the *PCElbowPlot* function was used to calculate the standard deviations of the principal components (PCs). PCs with standard deviation >3.5 were chosen as input parameters for nonlinear UMAP clustering analysis. Next, the *FindNeighbors* function was used to compute the k.param nearest neighbors, and BM cell clusters were identified by a shared nearest neighbor modularity optimization clustering algorithm implemented in the *FindClusters* function with resolution set at 0.4 (*Guo et al., 2019*; *Wang et al., 2021*; *Wen et al., 2021*).

## Analysis of potential receptor–ligand interactions using single-cell RNAseq datasets

The R package *NicheNet* was used to analyze the information about expression of cognate ligands and receptors to identify intercellular communication patterns between hematopoietic progenitors

and stromal cells as described (*Browaeys et al., 2020*). EMP and GMP cells were chosen as potential sources of receptors, whereas BM stromal cell types were chosen as potential sources of ligands. The background expression of genes was specified with default approach used in the *NicheNet* pipeline, and expressed genes were identified based on >10% detection in specific clusters. To identify ligand–receptor interactions between EMPs/GMPs and stromal cells, we selected the top 20 ligands predicted to drive hematopoietic cell differentiation based on the Pearson correlation coefficient between the ligand–receptor regulatory potential score of each ligand and the target indicator vector. Using the *NicheNet* pipeline, the Circos plots were generated to show common ligand–receptor interactions between EMPs/GMPs and stromal cells in the BM.

## FACS analysis

FACS analysis was performed using cells obtained from the BM and blood. Antibodies for FACS analysis are listed in *Supplementary file 3*. Immunostaining of cell suspensions were performed as described (*Bolte et al., 2017*; *Xia et al., 2015*). Identification of hematopoietic cell types based on multiple cell surface markers is described in *Bolte et al., 2020b*; *Pradhan et al., 2019*; *Ren et al., 2013*; *Ren et al., 2010*; *Sun et al., 2017*. To identify ESC-derived HSCs, we used GFP fluorescence and mouse-specific antibodies recognizing multiple cell surface antigens. First, ESC-derived GFP$^+$ BM cells were subdivided into Lin$^+$ and Lin$^-$ cell subsets. Second, we used Sca1 and CD117 (c-KIT) antibodies to identify Lin$^-$Sca1$^+$c-KIT$^+$ cells (LSKs). Third, CD150 and CD48 antibodies were used to identify ST-HSCs and LT-HSCs among LSKs. Stained cells were analyzed using a five-laser FACSAria II (BD Biosciences) (*Cai et al., 2016*; *Sun et al., 2021*).

## Histology and immunostaining

Frozen or paraffin-embedded sections of tissue samples were stained with hematoxylin and eosin (H&E) for histological evaluation (*Kalinichenko et al., 2002*) or to visualize GFP (*Ustiyan et al., 2018*; *Ustiyan et al., 2016*). Frozen sections from embryos were used for immunofluorescent staining as described (*Black et al., 2018*; *Ustiyan et al., 2012*; *Wang et al., 2010*). Primary antibodies for immunostaining are listed in *Supplementary file 3*. Secondary antibodies were conjugated with Alexa Fluor 488, Alexa Fluor 594, or Alexa Fluor 647 (Invitrogen and Jackson ImmunoResearch Laboratory) to visualize specific staining as described (*Bolte et al., 2012*; *Hoggatt et al., 2013*; *Milewski et al., 2017a*). DAPI (Vector Laboratory) was used to counterstain cell nuclei (*Milewski et al., 2017b*). Histological and immunofluorescent images were obtained using a Zeiss Axioplan2 microscope (Carl Zeiss Microimaging) as described (*Bolte et al., 2015*; *Kalin et al., 2008*; *Pradhan et al., 2016*).

## Statistical analysis

Statistical significance was determined using nonparametric Mann–Whitney *U*-test, one-way analysis of variance, and Student's *t*-test. Multiple means were compared using one-way analysis of variance with the post hoc Tukey test. p ≤0.05 was considered statistically significant. Data were presented as mean ± standard error of mean (SEM).

## Acknowledgements

We thank Mrs. Erika Smith for excellent secretarial support. This work was supported by NIH Grants HL141174 (to VVK), HL149631 (to VVK), and HL152973 (to VVK and TVK).

## Additional information

### Funding

| Funder | Grant reference number | Author |
| --- | --- | --- |
| National Heart, Lung, and Blood Institute | HL141174 | Vladimir V Kalinichenko |
| National Heart, Lung, and Blood Institute | HL149631 | Vladimir V Kalinichenko |

| Funder | Grant reference number | Author |
|---|---|---|
| National Heart, Lung, and Blood Institute | HL152973 | Vladimir V Kalinichenko |
| National Heart, Lung, and Blood Institute | HL158659 | Tanya V Kalin |

The funders had no role in study design, data collection, and interpretation, or the decision to submit the work for publication.

## Author contributions
Bingqiang Wen, Conceptualization, Resources, Data curation, Software, Formal analysis, Supervision, Validation, Investigation, Visualization, Methodology, Writing - original draft, Writing - review and editing; Guolun Wang, Resources, Data curation, Software, Formal analysis, Validation, Investigation, Visualization, Methodology, Writing - review and editing; Enhong Li, Data curation, Formal analysis, Validation, Visualization; Olena A Kolesnichenko, Resources, Investigation, Methodology, Writing - original draft, Writing - review and editing; Zhaowei Tu, Investigation, Methodology; Senad Divanovic, Investigation, Methodology, Writing - review and editing; Tanya V Kalin, Resources, Formal analysis, Funding acquisition, Investigation, Writing - original draft, Writing - review and editing; Vladimir V Kalinichenko, Conceptualization, Resources, Data curation, Supervision, Funding acquisition, Validation, Investigation, Methodology, Writing - original draft, Project administration, Writing - review and editing

## Author ORCIDs
Bingqiang Wen ⬤ http://orcid.org/0000-0001-8827-4820
Senad Divanovic ⬤ http://orcid.org/0000-0001-7538-0499
Vladimir V Kalinichenko ⬤ http://orcid.org/0000-0003-3438-2660

## Ethics
All animal studies were reviewed and approved by the Institutional Animal Care and Use Committee of the Cincinnati Children's Research Foundation (protocol # IACUC2016-0038).

## Decision letter and Author response
Decision letter https://doi.org/10.7554/eLife.74018.sa1
Author response https://doi.org/10.7554/eLife.74018.sa2

# Additional files

## Supplementary files
• Supplementary file 1. The number and percentage of hematopoietic bone marrow (BM) cells containing both mouse and rat mRNAs (hybrid cells).

• Supplementary file 2. The number of counts and features (genes) in six hybrid cells identified in mouse–rat chimera.

• Supplementary file 3. Antibodies used for flow cytometry (FC) and immunofluorescence (IF) staining.

• Transparent reporting form

• Source data 1. Excel spreadsheet containing quantitative data for *Figures 1, 5 and 7*.

## Data availability
Bone marrow single-cell RNA sequencing data have been deposited in GEO under accession number GSE184940.

The following dataset was generated:

| Author(s) | Year | Dataset title | Dataset URL | Database and Identifier |
|---|---|---|---|---|
| Wen B, Wang G, Kalinichenko VV | 2021 | The integrated single cell RNAseq analysis of bone marrow cells produced by mouse-mouse intraspecies blastocyst complementation and mouse-rat interspecies blastocyst complementation | https://www.ncbi.nlm.nih.gov/geo/query/acc.cgi?acc=GSE184940 | NCBI Gene Expression Omnibus, GSE184940 |

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
