## [Editor Report]

This work convincingly establishes a chimeric blastocyst complementation assay as a "bioreactor" to study the differentiation of mouse embryonic stem cells into hematopoietic lineages. The elegance of the approach lies in the use of GFP+ mouse embryonic stem cells that are implanted into a rat blastocyst, thus allowing for the tracking and phenotyping of the mouse-derived GFP+ hematopoietic cells in the post-natal rat. This is an important contribution that will be of interest to researchers in developmental biology and hematopoiesis.

---

## [Decision Letter]

**Decision letter after peer review:**

Thank you for sending your article entitled "in vivo Generation of Bone Marrow from Embryonic Stem Cells in Interspecies Chimeras" for peer review at *eLife*. Your article is being evaluated by 3 peer reviewers, and the evaluation is being overseen by a Reviewing Editor and Didier Stainier as the Senior Editor.

The reviewers were impressed by the chimera assay because it has the potential to uncover significant mechanisms of hematopoiesis and how specific niche interactions instruct hematopoiesis. However, the reviewers also felt that in its present form, the manuscript established the chimera model without showing its utility for gaining specific new mechanistic insights.

To leverage your blastocyst chimera model and derive important new insights, there would be the need for new analyses of the scRNA-seq data as well as new experimental studies that result in mechanistic insights. Your action plan for addressing these comments would help us reach a definitive decision.

*Reviewer #1 (Recommendations for the authors):*

Wen et al., establish the blastocyst complementation assay as a bioreactor; that helps direct the differentiation of mouse ESCs into the complete hematopoietic lineages. The elegance of the approach lies in the use of GFP+ mouse ESCs that are implanted into a rat blastocyst, thus allowing for the tracking and phenotyping of the mouse derived GFP+ hematopoietic cells in the post-natal rat. Single cell RNA-sequencing, flow cytometry and serial transplantation convincingly demonstrate that mouse ESCs differentiate into true HSCs as well as the more mature hematopoietic lineages. This further shows that the rat blastocyst can serve as a bioreactor which provides all the necessary cues to direct mouse ESC differentiation. The blastocyst complementation assay could therefore serve as a starting point to unravel the niche cues required for ESC differentiation into HSCs and hematopoietic progenitors as well generation of ex vivo bioreactors to generate and expand fully functional HSCs.

Key strengths:

1. The use of GFP+ ESCs is an elegant approach to track the fate of mouse ESCs.

2.The single cell RNA-seq analysis and flow cytometry data show the full spectrum of differentiation of mESCs.

3. Transplantation of mESC derived GFP+ cells nicely demonstrates that the generated HSCs are fully functional.

4. The described blastocyst complementation assay could be a useful platform for dissecting mechanisms of niche cues that direct ESC differentiation into hematopoietic lineages.

Key weaknesses:

1. A major feature of a "bioreactor" is the understanding and characterization of its key cellular components or molecules that drive its function. While the blastocyst complementation assay is very successful in generating fully functional HSCs and HSC progeny by the time cells are analyzed post-natally at P10, it is unclear how the blastocyst provides this information and, specifically, which cells or molecules in the blastocyst are essential for ESC differentiation into hematopoietic lineages.

2. The bioinformatic analysis of single cell RNA-seq data from GFP+ cells primarily uses general clustering and correlation analyses to assess similarity of mouse-rat chimera vs control. However, these analyses are performed at P10 and do not provide any clear insights or even suggestions as to how the blastocyst environment might have guided mouse ESCs to differentiation into hematopoietic lineages during the earlier developmental timepoints.

There are some key suggestions that would really help increase the impact of the work:

Can the single cell analysis be performed during one or more jey developmental timepoints after blastocyst complementation so that one could perhaps obtain insights into how the blastocyst begins to program ESCs during development?

Separating GFP+ and GFP- cells in the developing embryo at a timepoint when hematopoiesis occurs from a defined hematopoietic tissue such as the fetal liver could provide important insights into interactions between mESCs and the blastocyst role as a niche, as well as which niche components form from the GFP+ implanted ESCs.

Furthermore, a more in-depth analysis of predicted ligand-receptor interactions (using standard databases of ligand-receptor pairs) with inclusion of potential niche cells (stromal cells, endothelial cells, etc) that guide hematopoiesis in the scRNa-seq because they likely provide ligands for hematopoietic cells, and the inference or prediciton of transcription factor activities in the various cells during embryogenesis would also help inform how the blastocyst is driving the ESC differentiation towards hematopoietic cells.

Such experiments would leverage the elegance of the GFP+ mESC implantation and lineage tracing to identify specific cell types and pathways that one could modulate in order to understand novel mechanisms by which the "bioreactor" functions.

*Reviewer #2 (Recommendations for the authors):*

In the current study the authors utilize a well-developed system of interspecies chimaera to show conclusively that murine embryonic stem cells implanted in a rat blastocyst to produce viable mouse-rat chimaeras. This finding is strengthened by two observations: (1). The single-cell transcriptomic profile of the mouse-rat chimaeras closely matches that of the control mouse-mouse chimaeras for both the HSCs/progenitors as well as individual lineages. (2). The functional potential of mouse-rat chimaeras is tested by transplantation assays that show the chimaera derived cells to be capable of multi-lineage reconstitution. These two aspects of the study are both carefully conducted and supported the conclusion that the chimeric system, in the bone marrow niche, faithfully replicates the physiological and functional development of hematopoiesis.

While there are a few technical details that need to be addressed, the main critique of the study is the fact that characterization of hematopoiesis is limited to the bone marrow of newborn mouse-rat chimaeras. As the authors themselves admit, the blastocyst complementation system has already been shown to produce bone marrow hematopoietic cells in mouse-mouse chimaeras with added complexity of transgenic mutants (Hamanaka S et al., 2018), thus the observation that their mouse-rat chimaeras produce functional bone marrow HSCs is not inherently novel and primarily relies on the careful characterization of single cell transcriptional profiles to indicate that the mouse ESCs have produced transcriptionally authentic HSCs.

What the study needs in order to stand out and make ideal use of the mouse-rat chimeric system is a developmental perspective of how the murine ESCs are converted or incorporated into the embryo in order to produce definitive murine HSCs from the hemogenic endothelium of dorsal aorta. Or alternatively a better characterization, functional or transcriptional of the fetal GFP+ murine HSCs during maturation in the fetal liver.

It would be very enlightening to determine if the murine tissue must transition directly from the hemogenic endothelium to functional HSCs or if it can emerge denovo from non-GFP+ hemogenic endothelium. Since the cells are GFP-labeled and clearly identifiable from the rat tissue, this analysis can be performed by flow cytometry, single-cell RNAseq or even imaging and microscopy. It is not known when and how the chimeric system produces definitive HSCs in the embryos and the system that investigators have utilized to study the adult bone marrow is capable of answering this important question and thus producing a truly novel study.

The authors have done a commendable job characterizing the bone marrow contribution of the blastocyst complementation system in their mouse-rat chimaeras. Their results clearly show the amount and the expression profiles GFP+ murine HSCs and progenitors. However, on its own this characterization is not enough to merit publication as it is not mechanistically novel and makes use of previously established systems and only incrementally advances the field by using the chimeric system in an inter-species model.

The single cell RNAseq is informative and drives home the point that the murine ESCs produce the HSCs that are transcriptionally similar to mouse-mouse chimaeras, but the fact that the cells are capable of reconstitution in a transplant setting is already sufficient to establish the functional potential of the mouse HSCs in a rat chimeric bone marrow. Thus, a mechanistic and/or developmental approach is required to determine when and how the mouse ESCs produce the definitive HSCs in the chimeric fetus. This can already be performed with the system that the authors have in hand and only requires timed matings and chimeric embryo extraction and characterization of the fetal GFP+ cells either in the AGM region of the dorsal aorta during HSC emergence or the fetal liver during HSC expansion.

It would be very interesting to determine if the mouse HSCs emerge at the same rate as the rat ones or if there is a different selection during emergence and expansion in the fetal liver that produces the high frequency of GFP+ LT-HSCs seen in the bone marrow (Figure 1H). This integration and incorporation of the mouse cells in the chimeric system during embryo development would truly set the story apart and provide novel findings essential to our understanding of developmental hematopoiesis and chimeric model systems in general.

Specific comments:

In Fig,1, for both the legend and the text of the manuscript, the use of the term "control" and "mouse" is used interchangeably, but it is difficult to understand what it refers to. The comparison of a WT mouse to a GFP+ mouse-rat chimaera is not ideal as a control in this case because in a mouse presumably all the HSCs and progenitors are detectable, while in a mouse-rat chimaera only a subset of the cells are GFP+HSCs. This is very evident in Figure 1F where the "mouse" seems to have no GFP+ BM contribution in the BM while the chimaera has on average 25% contribution of GFP+ cells in the BM. It is not obvious why this is the case? A better control would be the assessment and comparison of the rat HSCs and progenitors in the mouse-rat BM of the chimeric mice.

In Figure 3 and Figure 4 how does the expression of the endogenous rat HSCs compare to that of the mouse HSCs in the chimaera. Presumably these two similar cell types from different species share a common bone marrow niche and thus are expected to be similar unless the presence of the mouse GFP+ HSCs and progenitors affects the endogenous rat cells.

Whole bone marrow transplants (with 500,000 GFP+ cells) are not the ideal means to establish cell autonomous functionality of the HSCs. For this purpose, phenotypic GFP+ HSCs, as shown in Figure 1E, should be sorted and transplanted into irradiated donors. Ideally, the transplant should be performed once again in secondary recipients to fully establish the functional potential and self-renewal capacity of the chimeric HSCs.

Figure 6A is not necessary since it is entirely subjective and has no values listed in any of the chosen gates. It should be moved to the supplement and Figure 5 should then be combined with Figure 6.

Figure 7A is not informative and should be removed from the manuscript. Instead, the authors should perform longitudinal sections of the bone (femur) and stain for H and E to show the effects of IR on the bone marrow microenvironment in the listed timepoints and treatments.

*Reviewer #3 (Recommendations for the authors):*

Rat-mouse interspecies chimeras have been used for many years and are useful to study the impact of various gene deficiencies in reconstituting different organs or tissues by mixing mutated and wild type cells. Here the authors have focused on the bone marrow and more particularly on the hematopoietic compartment.

The paper by Weng et al., aims at investigating the formation of the bone marrow with a focus on the hematopoietic compartment. They are using rat-mouse interspecies chimeras where mouse embryonic stem cells GFP-tagged are injected into rat blastocysts that subsequently develop into viable chimeric animals after reimplantation into pseudo-pregnant females. The authors sort the bone marrow cells from the chimeras on the basis of GFP and analyzed the hematopoietic cell populations with the standard flow cytometry approaches and reconstitution analysis in irradiated recipients. They conclude that a mouse-derived complete and functional hematopoietic hierarchy is present in the chimeric animals with an increase of short-term and long-term mouse hematopoietic stem cells compared to non-injected mice.

They analyzed the hematopoietic cell populations using single cell RNA sequencing and found weak or no difference compared to non-chimeric animals and they probed the reconstitution potential of the hematopoietic stem cell compartment. This is an interesting study that is essentially descriptive regarding the system of chimerism. The authors have however biased the analysis by focusing on the hematopoietic compartment and not on the entire bone marrow as they claim. Of note, the significant increase in the hematopoietic stem cell compartment observed in the chimeric context has not been analyzed. This is a key point of the study since it may help isolating factors to amplify hematopoietic stem cells. Thus despite an interesting design, this study falls short in finding key mechanisms of hematopoietic stem cell homeostasis and self-renewal.

Strengths

Exploring alternative approaches for the production of transplantable hematopoietic stem cells for therapeutic purposes.

Mastering interspecies rat-mouse chimeras to study the functioning of species-specific hematopoietic systems in a mixed bone marrow environment.

Weaknesses

Short, mostly descriptive study with few insights into how to improve hematopoietic stem cell manipulation and amplification.

Bias in the analysis by using only the hematopoietic fraction of the GFP+ population. The authors claim they are studying the bone marrow but they are in fact analyzing the hematopoietic compartment. The stromal cell compartment is totally lacking.

The single cell approach is biased since authors are artificially mixing different percentages of lin+ and lin- cells. The most interesting part of the work i.e., the amplification of short-term and long-term hematopoietic stem cells in chimeras should not be approached this way.

The long term reconstitution lacks secondary transplantation to show that hematopoietic stem cells are endowed with long-term reconstitution potential.

This is an interesting study that might give clues into how create and/or amplify hematopoietic stem cells.

Please focus on the entire bone marrow rather than on the hematopoietic cell compartment.

Analyze the transcriptome without introducing bias. The idea is to understand how the hematopoietic stem cell compartment is expanded in chimeras, not to show that all the hematopoietic cell lineages are present.

Analyze the stromal cell compartment that is complementary to the hematopoietic stem cell compartment.

Perform secondary transplantation to show that secondary reconstitution can take place.

Expand your analysis to cytokines and growth factors.

[Editors’ note: further revisions were suggested prior to acceptance, as described below.]

Thank you for resubmitting your work entitled "in vivo Generation of Bone Marrow from Embryonic Stem Cells in Interspecies Chimeras" for further consideration by *eLife*. Your revised article has been evaluated by Didier Stainier (Senior Editor) and a Reviewing Editor.

The manuscript has been improved but there are some remaining issues that need to be addressed, as outlined below:

1. The concern raised by Reviewer 2 about the functional potential of the GFP+ mouse BM HSCs from the chimeric mice.

2. Addressing the comments of Reviewer 3 about the presentation of specific figure panels to illustrate the key points of the study in a cohesive manner and the need for sufficient rationale + context.

It is likely that the questions can be addressed with existing data which needs to be prioritized (main figures versus supplement), presented in a cohesive manner, elaborated (experimental details) on and discussed more extensively (whenever there are limitations which affect the conclusions).

*Reviewer #1 (Recommendations for the authors):*

The authors have substantially expanded the manuscript through highly relevant new studies in response to the reviewer comments and have thereby increased the robustness and significance of their findings.

The significant improvements include the addition of more time points, the analysis of putative ligand receptor interactions and the secondary bone marrow transplantation studies.

The scRNA-seq data have been uploaded to GEO and are accessible to the public.

*Reviewer #2 (Recommendations for the authors):*

Overall manuscript is much improved, has broader appeal and now provides very intriguing findings on the developmental perspective of when HSCs emerge in a chimeric embryo as a potential timing regulation of LT-HSCs expansion in the fetal liver.

The authors were mostly responsive to the critiques and have sufficiently improved the novelty of the manuscript by performing experiments in the developing embryo.

However, there is one remaining critical concern regarding the functional potential of the GFP+ mouse BM HSCs from the chimaera. The authors say that they attempted to transplant sorted GFP+ stem cells but only about 198 LT-HSCs and ST-HSC were purified on average per chimaera BM. Furthermore, they say that these cells didn't provide enough hematopoietic support for a viable recipient mouse. This is not a valid response and raises a very big question about the cell intrinsic functional potential of the GFP+ mouse BM HSCs. Considering that 500,000 total cells provide reconstitution even long-term secondary transplant reconstitution, and with Figure 1H showing that there are chimaeras with as many as 40% GFP+ ST-HSCs and LT-HSCs, the ability to conduct a transplant with purified HSCs should be possible.

The authors did not have to test the average of GFP+ HSCs per chimaera, they simply had to show that some of the GFP+ cells were capable of cell autonomous long-term reconstitution. This brings to question the absolute number of mouse GFP+ HSCs in the chimaera bone marrow. Which should be determined from their flow cytometric analysis. The authors have to explain how they purified the ST- and LT-HSCs from the bone marrow and why 198 could not provide them with a viable recipient mouse when work from the Weissman and Morrison labs have shown that as few as 10 LT-HSCs (SLAM LSKs) can provide functional hematopoiesis in a transplant setting. Did the authors support the recipient mice with lineage+ splenocytes? What was the dose of irradiation administered? There has to be a better description of this experiment in the methods, in fact the methods as listed in the revision are very light on details of transplantation and lineage analysis.

So while the functional potential of chimeric stem cells has been established from the bulk bone marrow transplants, the lack of purified GFP+ mouse HSCs transplants limits the conclusions that the authors make with regard to calling these mouse HSCs functional in their experimental setting.

*Reviewer #3 (Recommendations for the authors):*

This reviewer thanks the authors for the revised version of their manuscript that shows significant improvement.

On the basis of this revised version, you will find enclosed my main comments for manuscript improvement. In general, I have the feeling that some of the figures are redundant or that the authors have trouble making a choice on what they want to show.

Figure 1: The authors examine the percentages of the ST-HSC and LT-HSC in WT mice vs chimeric animals. Ideally, a mouse/mouse chimeric control should also be included. In the Sca-Kit analysis (Fig1E); the authors mentioned a significant increase in the Sca-Kit compartment in chimeras vs WT animals. This is substantiated in Fig1G, although two groups of animals with distinct (high and low) percentages of Sca-Kit are visible but not substantiated in Figure 1E where the authors show close percentages between the two conditions. The same remark applies in the opposite direction for the percentages of LT vs ST HSC. I would recommend more carefully choosing the panels to illustrate their conclusions. The rationale of analyzing the animals at P10 is rather uncommon in hematology unless you want to analyze the forming bone marrow. P30 would have been more classical.

Figure 2: The design of this experiment is still puzzling for me and is not clearly indicated either in the text or in the legend. The authors have sorted GFP+ cells from the BM. Why is it necessary to enrich in committed hematopoietic progenitors since the BM already contains high numbers of these cells? Please clarify. The rationale of mixing mouse/mouse and mouse/rat GFP+ cells is not clear since the authors do not identify cells originating from one combination vs the other. At this point, instead of performing a scRNA-seq, a GFP+ FACS sorting followed by a multilineage hematopoietic analysis by flow cytometry i.e.myeloid, lymphoid, erythroid cells would have been more simple and direct. Could you please clarify the design of the experiment?

More generally the authors strongly emphasize on the transcriptome analysis. Several main figures are redundant and could be switched to supplementary ones e.g. Figure 3 and Figure 4 since these latter does not bring any specific insights into species-specific mechanisms.

Fig6 should be revised. The sections of the embryonic aorta are not informative regarding the endothelium and the hematopoietic cell production. Please provide images showing low magnification of the aorta and the surrounding structures and high magnification of the hematopoietic clusters.

Fig8: Multilineage analysis of the blood is interesting but the most informative piece of data comes from the BM analysis. Showing results at 8 days post-irradiation is not very informative since the animals are in aplasia.

---

## [Author Response]

Reviewer #1 (Recommendations for the authors):Wen et al., establish the blastocyst complementation assay as a bioreactor; that helps direct the differentiation of mouse ESCs into the complete hematopoietic lineages. The elegance of the approach lies in the use of GFP+ mouse ESCs that are implanted into a rat blastocyst, thus allowing for the tracking and phenotyping of the mouse derived GFP+ hematopoietic cells in the post-natal rat. Single cell RNA-sequencing, flow cytometry and serial transplantation convincingly demonstrate that mouse ESCs differentiate into true HSCs as well as the more mature hematopoietic lineages. This further shows that the rat blastocyst can serve as a bioreactor which provides all the necessary cues to direct mouse ESC differentiation. The blastocyst complementation assay could therefore serve as a starting point to unravel the niche cues required for ESC differentiation into HSCs and hematopoietic progenitors as well generation of ex vivo bioreactors to generate and expand fully functional HSCs.

We would like to thank the Reviewer for positive assessment of our manuscript and for recognizing future potential of our work.

Key strengths:1. The use of GFP+ ESCs is an elegant approach to track the fate of mouse ESCs.2.The single cell RNA-seq analysis and flow cytometry data show the full spectrum of differentiation of mESCs.3. Transplantation of mESC derived GFP+ cells nicely demonstrates that the generated HSCs are fully functional.4. The described blastocyst complementation assay could be a useful platform for dissecting mechanisms of niche cues that direct ESC differentiation into hematopoietic lineages.

We would like to thank the Reviewer for summarizing key strengths of our manuscript.

Key weaknesses:1. A major feature of a "bioreactor" is the understanding and characterization of its key cellular components or molecules that drive its function. While the blastocyst complementation assay is very successful in generating fully functional HSCs and HSC progeny by the time cells are analyzed post-natally at P10, it is unclear how the blastocyst provides this information and, specifically, which cells or molecules in the blastocyst are essential for ESC differentiation into hematopoietic lineages.

We agree. We provided additional experimental data in new Figures 4, 5, 6, and 7D, and new Figure 4 —figure supplement 1A-D, Figure 5 —figure supplements 1 and 2, Figure 6 —figure supplements 1A-C and 2, Figure 8 —figure supplement 6A-E. that characterize both hematopoietic and stromal cellular components in the chimeric bone marrow and analyze receptor-ligand interactions between hematopoietic and stromal cells in the chimeras. We also characterized the development of donor hematopoietic progenitors during early neonatal and embryonic time points in the bone marrow, dorsal aorta and liver of the mouse-rat chimeras. Specific new findings are summarized below in recommendations for the authors.

2. The bioinformatic analysis of single cell RNA-seq data from GFP+ cells primarily uses general clustering and correlation analyses to assess similarity of mouse-rat chimera vs control. However, these analyses are performed at P10 and do not provide any clear insights or even suggestions as to how the blastocyst environment might have guided mouse ESCs to differentiation into hematopoietic lineages during the earlier developmental timepoints.

We agree. We provided additional bioinformatic analysis of new single-cell RNA sequencing of P5 bone marrow in the mouse, rat and mouse-rat chimera which includes both hematopoietic and stromal cells of mouse (GFP+) and rat origin (GFP-) (new Figure 4 and Figure 4 —figure supplement 1A-D). We also expanded the bioinformatic analysis to analysis of receptor-ligand interactions between hematopoietic and stromal cells in the rat and mouse compartment of the chimeras (Figure 5 and Figure 5 —figure supplements 1 and 2). Finally, we characterized the contribution of mouse ESCs to HSCs during embryonic development of mouse-rat chimeras by providing new immunostaining of dorsal aorta (new Figure 6A) and FACS analysis of the fetal liver (new Figure 6B and Figure 6 —figure supplements 1A-C and 2).

There are some key suggestions that would really help increase the impact of the work:Can the single cell analysis be performed during one or more jey developmental timepoints after blastocyst complementation so that one could perhaps obtain insights into how the blastocyst begins to program ESCs during development?

We followed up on this excellent suggestion. During these experiments, we determined that the number of cells from the bone marrow compartment of embryos is very limited and insufficient to perform single-cell RNA seq. Therefore, we performed an additional single-cell RNA seq using the early neonatal time-point (P5). Analysis of receptor-ligand interactions between hematopoietic and stromal cells in the rat and mouse compartment of the chimeras was performed and provided in (Figure 5 and Figure 5 —figure supplements 1 and 2). Furthermore, we used developmental time-points E12.5 and E15.5 from mouse-rat chimeras to characterize the contribution of mouse ESCs to HSCs by immunostaining (new Figure 6) and FACS analysis (new Figure 6B and Figure 6 —figure supplements 1A-C and 2).

Overall, these new data demonstrate that chimeric HSCs develop earlier and more efficiently from donor mouse ESCs compared to endogenous rat ESCs. Since mouse embryos develop faster compared to rat embryos by approximately 1.5 days (new references: Farrington-Rock et al., 2008; Marcela et al., 2012; Takahashi and Osumi, 2005; Torres et al., 2008; and new Figure 6 —figure supplement 1A-C), it is possible that mouse ESC-derived progenitor cells migrate faster into developing hematopoietic niches in the mouse-rat chimeras, leading to preferential development of HSCs from cells of mouse origin and contributing to increased numbers of mouse-derived hematopoietic progenitors in the bone marrow of mouse-rat chimeras. We have included this possibility in the Discussion section (pages 14-15).

Separating GFP+ and GFP- cells in the developing embryo at a timepoint when hematopoiesis occurs from a defined hematopoietic tissue such as the fetal liver could provide important insights into interactions between mESCs and the blastocyst role as a niche, as well as which niche components form from the GFP+ implanted ESCs.

We agree. As mentioned above, we analyzed separately GFP+ and GFP- hematopoietic cells and their interactions with stromal cells in P5 bone marrow (new Figures 4 and 5, and new Figure 4 —figure supplement 1A-D and Figure 5 —figure supplements 1 and 2). The bone marrow is more relevant to our study than embryonic liver tissue because of fundamental differences in hematopoietic niche components in the bone marrow compared to the liver, which contains hepatocytes, hepatoblasts, bile duct epithelial cells, stellate cells and other liver-specific cell types that are absent in cellular niches of the bone marrow.

Furthermore, a more in-depth analysis of predicted ligand-receptor interactions (using standard databases of ligand-receptor pairs) with inclusion of potential niche cells (stromal cells, endothelial cells, etc) that guide hematopoiesis in the scRNa-seq because they likely provide ligands for hematopoietic cells, and the inference or prediciton of transcription factor activities in the various cells during embryogenesis would also help inform how the blastocyst is driving the ESC differentiation towards hematopoietic cells.

We agree. As requested by the Reviewer, we provided in-depth analysis of predicted receptor-ligand interactions between hematopoietic and niche cells (including stromal cells, endothelial cells, chondrocytes, etc.) in the rat and mouse compartment of the chimeric bone marrow (new Figure 5 —figure supplement 1). We also compared gene expression signatures and the expression of key ligands and their receptors between mouse and rat cells in the chimeric bone marrow (new Figure 5 —figure supplement 2). These new data were incorporated into the revised manuscript (page 10).

Such experiments would leverage the elegance of the GFP+ mESC implantation and lineage tracing to identify specific cell types and pathways that one could modulate in order to understand novel mechanisms by which the "bioreactor" functions.

We agree. Using the GFP+ mESC implantation into rat blastocysts followed by the lineage tracing of mESC-derived cells, we identified several specific signaling pathways between hematopoietic progenitors and stromal cells which can be involved in the function of the “bioreactor”. Focusing on interaction between erythroid-myeloid progenitor cells (EMPs) and stromal cells in the bone marrow, we identified *Cxcl12-Cxcr4, Lama2-Itga6, App-Itga6* and *Comp-Cd47* receptor-ligand pairs as major predicted interactions between EMPs and endothelial cells, fibroblasts and chondrocytes (new Figure 5). The major signaling pathways between granulocyte-myeloid progenitor cells (GMPs) and stromal cells were *Col1a1-Cd44, App-Il18rap* and *Cxcl12-Cxcr4* (new Figure 5 —figure supplement 1). We included these data in the Results section (page 10). We also provided new references (Baccin et al., 2020; Singh et al., 2020; Sugiyama et al., 2006; Kiratipaiboon et al., 2020; Mk et al., 2019; Rock et al., 2010; Strelnikov et al., 2021; Yang et al., 2017), which support the potential importance of these pathways in cell signaling between hematopoietic and stromal cells. Finally, we modified the Discussion section to suggest that these pathways can be targeted to modulate the development of donor ESC-derived hematopoietic progenitor cells in the “bioreactor” (page 15).

Reviewer #2 (Recommendations for the authors):In the current study the authors utilize a well-developed system of interspecies chimaera to show conclusively that murine embryonic stem cells implanted in a rat blastocyst to produce viable mouse-rat chimaeras. This finding is strengthened by two observations: (1). The single-cell transcriptomic profile of the mouse-rat chimaeras closely matches that of the control mouse-mouse chimaeras for both the HSCs/progenitors as well as individual lineages. (2). The functional potential of mouse-rat chimaeras is tested by transplantation assays that show the chimaera derived cells to be capable of multi-lineage reconstitution. These two aspects of the study are both carefully conducted and supported the conclusion that the chimeric system, in the bone marrow niche, faithfully replicates the physiological and functional development of hematopoiesis.

We would like to thank the Reviewer for positive assessment of our manuscript and for the comment that our interspecies chimeric system replicates the physiological and functional development of hematopoiesis.

While there are a few technical details that need to be addressed, the main critique of the study is the fact that characterization of hematopoiesis is limited to the bone marrow of newborn mouse-rat chimaeras. As the authors themselves admit, the blastocyst complementation system has already been shown to produce bone marrow hematopoietic cells in mouse-mouse chimaeras with added complexity of transgenic mutants (Hamanaka S et al., 2018), thus the observation that their mouse-rat chimaeras produce functional bone marrow HSCs is not inherently novel and primarily relies on the careful characterization of single cell transcriptional profiles to indicate that the mouse ESCs have produced transcriptionally authentic HSCs.

To increase the novelty of our study, we performed additional single-cell RNA sequencing experiments to compare cells in the bone marrow of the mouse, rat, and mouse-rat chimeras, including both hematopoietic and niche cells (endothelial cells, chondrocytes, fibroblasts and myofibroblasts) of mouse (GFP+) and rat origin (GFP-) (new Figure 4 and Figure 4 —figure supplement 1A-D). Furthermore, we provided bioinformatic analysis of receptor-ligand interactions and identified signaling pathways between hematopoietic progenitors and stromal cells in both the rat and mouse BM compartments of interspecies mouse-rat chimeras (new Figure 5 and Figure 5 —figure supplements 1 and 2). Such detailed studies have not been conducted in the bone marrow of any interspecies chimeras, significantly increasing the novelty of our manuscript.

What the study needs in order to stand out and make ideal use of the mouse-rat chimeric system is a developmental perspective of how the murine ESCs are converted or incorporated into the embryo in order to produce definitive murine HSCs from the hemogenic endothelium of dorsal aorta. Or alternatively a better characterization, functional or transcriptional of the fetal GFP+ murine HSCs during maturation in the fetal liver.

We agree. As the Reviewer suggested, we characterized the contribution of mouse ESCs to HSCs during embryonic development of mouse-rat chimeras by providing new immunostaining of dorsal aorta for RUNX1 and FLK1 (new Figure 6A) and FACS analysis of the fetal liver (new Figure 6B and Figure 6 —figure supplement 2). Based on co-expression of RUNX1 transcription factor and FLK1 (VEGF receptor 2), hemogenic endothelium in chimeric dorsal aorta develops earlier from donor mouse ESCs compared to endogenous rat ESCs (Figure 6A). Furthermore, we used FACS analysis of fetal E15.5 livers for CD117, Sca1, CD48 and CD150 in Lin^–^ cells to show an increase in the number of fetal murine hematopoietic progenitor cells (Figure 6B and Figure 6 —figure supplement 2). All these data are consistent with increased numbers of murine HSCs in the bone marrow of mouse-rat chimeras (Figure 4).

It would be very enlightening to determine if the murine tissue must transition directly from the hemogenic endothelium to functional HSCs or if it can emerge denovo from non-GFP+ hemogenic endothelium. Since the cells are GFP-labeled and clearly identifiable from the rat tissue, this analysis can be performed by flow cytometry, single-cell RNAseq or even imaging and microscopy. It is not known when and how the chimeric system produces definitive HSCs in the embryos and the system that investigators have utilized to study the adult bone marrow is capable of answering this important question and thus producing a truly novel study.

We agree. Based on new FACS and immunostaining data described above, it is likely that donor mouse ESC-derived hemogenic endothelium undergoes direct transition to functional HSCs in the fetal liver, whereas rat-derived (non-GFP+) hemogenic endothelium can be a source of rat HSCs. We acknowledged this possibility in the Discussion section (pages 14-15).

The average gestational time in C57Bl/6 mice is 19 days, whereas the gestational time in CD-1 rats is 21 days. Since mouse embryos develop faster compared to rat embryos by approximately 1.5 days (new references Farrington-Rock et al., 2008; Marcela et al., 2012; Takahashi and Osumi, 2005; Torres et al., 2008, and new Figure 6 —figure supplement 1), it is possible that mouse ESC-derived cells migrate faster and colonize the developing hematopoietic niches in the mouse-rat chimeras, leading to a “competing advantage” and preferential development of murine HSCs compared to rat HSCs. These data are consistent with increased numbers of mouse ESC-derived hematopoietic progenitors in the dorsal aorta, fetal liver and bone marrow of mouse-rat chimeras. Our data suggest that using donor ESCs from species with less gestational time in interspecies “bioreactors” can lead to larger quantities of ESC-derived hematopoietic progenitors in the chimeric bone marrow. We included this statement in the Discussion section (page 15).

The authors have done a commendable job characterizing the bone marrow contribution of the blastocyst complementation system in their mouse-rat chimaeras. Their results clearly show the amount and the expression profiles GFP+ murine HSCs and progenitors. However, on its own this characterization is not enough to merit publication as it is not mechanistically novel and makes use of previously established systems and only incrementally advances the field by using the chimeric system in an inter-species model.The single cell RNAseq is informative and drives home the point that the murine ESCs produce the HSCs that are transcriptionally similar to mouse-mouse chimaeras, but the fact that the cells are capable of reconstitution in a transplant setting is already sufficient to establish the functional potential of the mouse HSCs in a rat chimeric bone marrow. Thus, a mechanistic and/or developmental approach is required to determine when and how the mouse ESCs produce the definitive HSCs in the chimeric fetus. This can already be performed with the system that the authors have in hand and only requires timed matings and chimeric embryo extraction and characterization of the fetal GFP+ cells either in the AGM region of the dorsal aorta during HSC emergence or the fetal liver during HSC expansion.

We agree. As suggested by the Reviewer we performed timed mating, chimeric embryo extraction, and characterization of the fetal GFP^+^ hemogenic endothelial cells in the dorsal aorta using immunostaining for RUNX1 and FLK1. We also performed FACS analysis of E15.5 livers for GFP^+^ HSCs. These data are provided in new Figure 6A-B and Figure 6 —figure supplement 2. Furthermore, we provided new single-cell RNA seq data of early postnatal (P5) bone marrow to characterize signaling pathways between stromal and hematopoietic compartments in donor (murine) and recipient (rat) cells in chimeric bone marrow (new Figures 4A-C and 5, and new Figure 4 —figure supplement 1A-D and Figure 5 —figure supplements 1 and 2). All these data were incorporated into the Results and Discussion sections of the revised manuscript (pages 9-11 and 14-15).

It would be very interesting to determine if the mouse HSCs emerge at the same rate as the rat ones or if there is a different selection during emergence and expansion in the fetal liver that produces the high frequency of GFP+ LT-HSCs seen in the bone marrow (Figure 1H). This integration and incorporation of the mouse cells in the chimeric system during embryo development would truly set the story apart and provide novel findings essential to our understanding of developmental hematopoiesis and chimeric model systems in general.

We agree. Our new data demonstrate that the mouse HSCs emerge faster compared to the rat ones in the dorsal aorta of mouse-rat chimeras (new Figure 6A). We also observed increased numbers of GFP+ murine ST-HSCs during expansion in the fetal E15.5 liver (new Figure 6 —figure supplement 2). In contrast, the number of LT-HSCs in the fetal liver was unchanged. It is possible that the expansion of murine LT-HSCs in mouse-rat chimeras occurs after E15.5, leading to the increased percentage of these cells in the chimeric bone marrow (Figure 1H). We hope that our new findings increase novelty of our manuscript and improve our understanding of developmental hematopoiesis in chimeric model system.

Specific comments:In Fig,1, for both the legend and the text of the manuscript, the use of the term "control" and "mouse" is used interchangeably, but it is difficult to understand what it refers to. The comparison of a WT mouse to a GFP+ mouse-rat chimaera is not ideal as a control in this case because in a mouse presumably all the HSCs and progenitors are detectable, while in a mouse-rat chimaera only a subset of the cells are GFP+HSCs. This is very evident in Figure 1F where the "mouse" seems to have no GFP+ BM contribution in the BM while the chimaera has on average 25% contribution of GFP+ cells in the BM. It is not obvious why this is the case? A better control would be the assessment and comparison of the rat HSCs and progenitors in the mouse-rat BM of the chimeric mice.

We agree. We provided new single-cell RNAseq of P5 bone marrows in which we directly compared mouse and rat hematopoietic progenitors in the mouse-rat chimeras. We also compared these cells to “normal BM cells” of rats and mice of the same age (new Figures 4 and 5, new Figure 4 —figure supplement 1A-D, and Figure 5 —figure supplements 1 and 2). Different contributions of mouse and rat cells in the mouse-rat chimeras were provided in new Figure 4C of the revised manuscript.

Unfortunately, Flow cytometry cannot be used to distinguish between mouse and rat cells in the chimeras because rat-specific Abs (which do not recognize mouse antigens) to most of cell surface markers are not commercially available. We provide Figure 1 —figure supplement 1A-B to show that mouse-specific Abs used in Figure 1 recognize only mouse but not rat cell surface molecules.

In Figure 3 and Figure 4 how does the expression of the endogenous rat HSCs compare to that of the mouse HSCs in the chimaera. Presumably these two similar cell types from different species share a common bone marrow niche and thus are expected to be similar unless the presence of the mouse GFP+ HSCs and progenitors affects the endogenous rat cells.

We agree. New single-cell RNAseq dataset from P5 chimeric BM was used to address this question. We provided new Figure 4 —figure supplement 1A-D in which we compared gene expression of endogenous rat hematopoietic and stromal cells with that of the mouse ESC-derived cells in the mouse-rat chimeras. Expression signatures and potential ligand-receptor interactions in mouse and rat cells were remarkably similar (new Figure 4 —figure supplement 1A-D, new Figure 5, and Figure 5 —figure supplements 1 and 2).

Whole bone marrow transplants (with 500,000 GFP+ cells) are not the ideal means to establish cell autonomous functionality of the HSCs. For this purpose, phenotypic GFP+ HSCs, as shown in Figure 1E, should be sorted and transplanted into irradiated donors. Ideally, the transplant should be performed once again in secondary recipients to fully establish the functional potential and self-renewal capacity of the chimeric HSCs.

As suggested by the Reviewer, we have performed this experiment by transplanting FACS-sorted, GFP^+^ HSCs (ST-HSC/ LT-HSC cell mixture) into irradiated recipient mice. The average number of transplanted GFP^+^ HSCs per mouse was 194 cells. In our experimental conditions, this number of HSCs was insufficient to rescue lethally irradiated mice, possibly, due to requirements for other progenitor and differentiated hematopoietic cells in the bone marrow to replace the loss of vast majority of white blood cells in the acute stage after the high-dose irradiation (Figure 8B, please see WBC numbers in the peripheral blood 8 days after irradiation).

To establish the functional potential and self-renewal capacity of the chimeric GFP^+^ HSCs, we performed the second bone marrow transplantation in secondary recipients and these new data are provided in Figure 8 —figure supplement 6A-E. The secondary transplantation of mouse ESC-derived bone marrow from the mouse-rat chimeras rescued irradiated mice (new Figure 8 —figure supplement 6B-C) and resulted in long-term engraftment into hematopoietic cell lineages of the peripheral blood and bone marrow (new Figure 8 —figure supplement 6D-E). These new data were incorporated into the revised manuscript (page 13).

Figure 6A is not necessary since it is entirely subjective and has no values listed in any of the chosen gates. It should be moved to the supplement and Figure 5 should then be combined with Figure 6.

We agree. As suggested by the Reviewer, we moved old Figure 6A to the supplement (new Figure 8 —figure supplement 1) and then combined the remaining panels of old Figures 6 and 7 into new Figure 8.

Figure 7A is not informative and should be removed from the manuscript. Instead, the authors should perform longitudinal sections of the bone (femur) and stain for H and E to show the effects of IR on the bone marrow microenvironment in the listed timepoints and treatments.

We agree. Figure 7A was removed from main figures and placed in the supplement (new Figure 8 —figure supplement 4). As requested by the Reviewer, we provided H and E-stained longitudinal sections of the femur in new Figure 7D. We also provided GFP images of frozen BM tissue sections to demonstrate the abundance of mouse ESC-derived (GFP+) cells in the BM compartment of transplanted mice (new Figure 7D, bottom panels).

Reviewer #3 (Recommendations for the authors):Rat-mouse interspecies chimeras have been used for many years and are useful to study the impact of various gene deficiencies in reconstituting different organs or tissues by mixing mutated and wild type cells. Here the authors have focused on the bone marrow and more particularly on the hematopoietic compartment.The paper by Weng et al., aims at investigating the formation of the bone marrow with a focus on the hematopoietic compartment. They are using rat-mouse interspecies chimeras where mouse embryonic stem cells GFP-tagged are injected into rat blastocysts that subsequently develop into viable chimeric animals after reimplantation into pseudo-pregnant females. The authors sort the bone marrow cells from the chimeras on the basis of GFP and analyzed the hematopoietic cell populations with the standard flow cytometry approaches and reconstitution analysis in irradiated recipients. They conclude that a mouse-derived complete and functional hematopoietic hierarchy is present in the chimeric animals with an increase of short-term and long-term mouse hematopoietic stem cells compared to non-injected mice.They analyzed the hematopoietic cell populations using single cell RNA sequencing and found weak or no difference compared to non-chimeric animals and they probed the reconstitution potential of the hematopoietic stem cell compartment.

We would like to thank the Reviewer for summarizing the results and conclusions of our manuscript.

This is an interesting study that is essentially descriptive regarding the system of chimerism. The authors have however biased the analysis by focusing on the hematopoietic compartment and not on the entire bone marrow as they claim. Of note, the significant increase in the hematopoietic stem cell compartment observed in the chimeric context has not been analyzed. This is a key point of the study since it may help isolating factors to amplify hematopoietic stem cells. Thus despite an interesting design, this study falls short in finding key mechanisms of hematopoietic stem cell homeostasis and self-renewal.

Based on constructive comments of the reviewers, we provided additional experimental data in new Figures 4, 5, 6, and 7D, and Figure 4 —figure supplement 1A-D, Figure 5 —figure supplements 1 and 2, Figure 6 —figure supplements 1A-C and 2, Figure 8 —figure supplement 6A-E, which characterize both hematopoietic and stromal cellular components in the chimeric bone marrow and analyze receptor-ligand interactions between hematopoietic and stromal cells in the chimeras. Our new studies identified *Cxcl12-Cxcr4, Lama2-Itga6, App-Itga6*, *Comp-Cd47, Col1a1-Cd44* and *App-Il18rap* signaling pathways between hematopoietic progenitors and stromal cells (new Figure 5 and Figure 5 —figure supplements 1 and 2). It is possible that modulating these pathways may help to amplify hematopoietic progenitor cells in the interspecies bone marrow. We also performed additional studies to characterize the development of donor hematopoietic progenitor cells during embryonic and early neonatal (P5) time points using the bone marrow, dorsal aorta, and fetal livers from mouse-rat chimeras (new Figures 4, 5 and 6, and Figure 4 —figure supplement 1A-D, Figure 5 —figure supplements 1 and 2, Figure 6 —figure supplements 1A-C and 2). Altogether, our new data provide important information about signaling and developmental mechanisms that lead to formation of the BM compartment in interspecies chimeras.

StrengthsExploring alternative approaches for the production of transplantable hematopoietic stem cells for therapeutic purposes.Mastering interspecies rat-mouse chimeras to study the functioning of species-specific hematopoietic systems in a mixed bone marrow environment.

We would like to thank the Reviewer for summarizing key strengths of our manuscript.

WeaknessesShort, mostly descriptive study with few insights into how to improve hematopoietic stem cell manipulation and amplification.Bias in the analysis by using only the hematopoietic fraction of the GFP+ population. The authors claim they are studying the bone marrow but they are in fact analyzing the hematopoietic compartment. The stromal cell compartment is totally lacking.

We performed additional experiments and provided new experimental data in which we used single-cell RNAseq to characterize both hematopoietic and stromal cellular components in the chimeric bone marrow (new Figures 4 and 5, and Figure 4 —figure supplement 1A-D, Figure 5 —figure supplements 1 and 2).

The single cell approach is biased since authors are artificially mixing different percentages of lin+ and lin- cells. The most interesting part of the work i.e., the amplification of short-term and long-term hematopoietic stem cells in chimeras should not be approached this way.

We would also like to point out that the mixing of different cell types was done because we followed the previously published single cell RNAseq protocol (Baccin et al., 2020, Nat Cell Biol), which allows to obtain sufficient numbers of rare hematopoietic progenitor cells for bioinformatic analysis.

In the revised manuscript, we provided new single-cell RNA sequencing datasets which were produced without mixing of Lin^+^ and Lin^–^ cells (new Figures 4 and 5, and Figure 4 —figure supplement 1A-D, Figure 5 —figure supplements 1 and 2). These new datasets were integrated with the old single cell RNAseq data in the revised manuscript.

The long term reconstitution lacks secondary transplantation to show that hematopoietic stem cells are endowed with long-term reconstitution potential.

We agree. As requested by the Reviewer, we performed the secondary bone marrow transplantation and found that mouse ESC-derived hematopoietic stem cells are endowed with long-term reconstitution potential (new Figure 8 —figure supplement 6A-E).

Reviewer #3 (Recommendations for the authors):This is an interesting study that might give clues into how create and/or amplify hematopoietic stem cells.Please focus on the entire bone marrow rather than on the hematopoietic cell compartment.

We agree. We provided additional experimental data in new Figures 4 and 5, and Figure 4 —figure supplement 1A-D, Figure 5 —figure supplements 1 and 2 that characterize both hematopoietic and stromal cellular components in the chimeric bone marrow.

Analyze the transcriptome without introducing bias. The idea is to understand how the hematopoietic stem cell compartment is expanded in chimeras, not to show that all the hematopoietic cell lineages are present.

We agree. We provided new single-cell RNAseq datasets of the bone marrow without mixing of Lin^+^ and Lin^–^ cells (new Figures 4 and 5, and Figure 4 —figure supplement 1A-D, Figure 5 —figure supplements 1 and 2) which were integrated in the revised manuscript. We also performed the receptor-ligand analysis between hematopoietic and stromal cells to highlight the potential significance of *Cxcl12-Cxcr4, Lama2-Itga6, App-Itga6*, *Comp-Cd47, Col1a1-Cd44* and *App-Il18rap* signaling pathways in expansion of hematopoietic progenitor cells in interspecies chimeras. Finally, we performed analysis of hemogenic endothelium and HSCs in the dorsal aorta and fetal liver and found that mouse cells faster colonize the developing hematopoietic niches in the mouse-rat chimeras (new Figure 6A-B and Figure 6 —figure supplements 1A-C and 2), possibly, leading to a “competing advantage” and preferential development of murine HSCs compared to rat HSCs.

Analyze the stromal cell compartment that is complementary to the hematopoietic stem cell compartment.

We agree. The analysis of the stromal BM compartment was performed using single-cell RNAseq. These data were included in the revised manuscript (new Figures 4 and 5, and Figure 4 —figure supplement 1A-D, Figure 5 —figure supplements 1 and 2).

Perform secondary transplantation to show that secondary reconstitution can take place.

We agree. We provided these data in new Figure 8 —figure supplement 6A-E.

Expand your analysis to cytokines and growth factors.

We agree. We used receptor-ligand analysis between hematopoietic and stromal cells to identify several signaling molecules and their receptors which can be involved in expansion of mouse hematopoietic progenitor cells in interspecies chimeras (new Figure 5 and Figure 5 —figure supplements 1 and 2).

[Editors’ note: further revisions were suggested prior to acceptance, as described below.]

The manuscript has been improved but there are some remaining issues that need to be addressed, as outlined below:

We would like to thank the Editor for handling the review of our manuscript and for providing constructive comments.

1. The concern raised by Reviewer 2 about the functional potential of the GFP+ mouse BM HSCs from the chimeric mice.

We addressed the concern of the Reviewer 2 by modifying our manuscript to avoid statements that “mouse HSCs are fully functional” in our experimental setting. Instead, we used the words “the bone marrow from mouse-rat chimeras was functional”. This statement is fully supported by our studies because (1) the bone marrow from mouse-rat chimers rescued lethally irradiated mice and restored all hematopoietic cell lineages in the peripheral blood and bone marrow after the primary and secondary transplantation; and (2) the bone marrow from mouse-rat chimers contains all ESC-derived hematopoietic progenitors and stromal cell subsets that were detected in normal mice and rats.

Furthermore, we modified the Discussion section to acknowledge a limitation of our study about HSC functional potential as follows (page 16):

“…One of the limitations of our studies is that the functional potential of chimeric HSCs was established from whole BM transplants and not from transplantation of purified HSCs. While these experiments are technically challenging, transplantation of FACS-sorted donor HSCs into lethally irradiated mice will be needed in our future studies to investigate whether chimeric HSCs are fully functional to restore all hematopoietic cell lineages after irradiation.”

2. Addressing the comments of Reviewer 3 about the presentation of specific figure panels to illustrate the key points of the study in a cohesive manner and the need for sufficient rationale + context.

We agree. As suggested by the Reviewer 3, we moved Figure 3 into the supplement, provided low magnification images of embryos in Figure 5A, replaced several flow cytometry panels in Figure 1E, provided new Figure 1I showing the total numbers of HSCs in chimeras, and included additional rationale and experimental details for single cell RNA-seq experiments. Specific points are addressed below in the answer to reviewer’s critique.

It is likely that the questions can be addressed with existing data which needs to be prioritized (main figures versus supplement), presented in a cohesive manner, elaborated (experimental details) on and discussed more extensively (whenever there are limitations which affect the conclusions).

We agree. Based on reviewer’s comments, we revised the presentation of experimental data in the main figures and supplements (main Figures1, 2 and 5, and Figure 2 —figure supplements 5, 6 and 7), provided additional experimental details in the Methods section (pages 18-20) and provided an additional discussion about limitations of our studies as related to transplantation of purified HSCs to irradiated mice (page 16).

Reviewer #1 (Recommendations for the authors):The authors have substantially expanded the manuscript through highly relevant new studies in response to the reviewer comments and have thereby increased the robustness and significance of their findings.The significant improvements include the addition of more time points, the analysis of putative ligand receptor interactions and the secondary bone marrow transplantation studies.The scRNA-seq data have been uploaded to GEO and are accessible to the public.

We would like to thank the Reviewer for summarizing the improvements in our revised manuscript and for acknowledging the robustness and significance of our findings.

Reviewer #2 (Recommendations for the authors):Overall manuscript is much improved, has broader appeal and now provides very intriguing findings on the developmental perspective of when HSCs emerge in a chimeric embryo as a potential timing regulation of LT-HSCs expansion in the fetal liver.The authors were mostly responsive to the critiques and have sufficiently improved the novelty of the manuscript by performing experiments in the developing embryo.

We would like to thank the Reviewer for acknowledging the improvements and novelty of our revised manuscript.

However, there is one remaining critical concern regarding the functional potential of the GFP+ mouse BM HSCs from the chimaera. The authors say that they attempted to transplant sorted GFP+ stem cells but only about 198 LT-HSCs and ST-HSC were purified on average per chimaera BM. Furthermore, they say that these cells didn't provide enough hematopoietic support for a viable recipient mouse. This is not a valid response and raises a very big question about the cell intrinsic functional potential of the GFP+ mouse BM HSCs. Considering that 500,000 total cells provide reconstitution even long-term secondary transplant reconstitution, and with Figure 1H showing that there are chimaeras with as many as 40% GFP+ ST-HSCs and LT-HSCs, the ability to conduct a transplant with purified HSCs should be possible.

We understand the Reviewer concern and we have tried to perform this experiment during the revision process which took approximately 9 months. Such experiments are time-consuming because the chimeras must be created before the transplantation. It is likely, that our inability to rescue lethally irradiated mice with FACS-sorted GFP+ HSCs can be explained by technical difficulties. Since HSC are rare, a long time of cell sorting can lead to decreased cell viability which directly affects the transplantation potential of HSCs. As the Reviewer noted, our Figure 1H shows that some chimeras have as many as 40% HSCs. However, this number of HSCs is among LSKs that account for minor percentage of Lin-negative cells in the bone marrow (Figure 1E and 1G). Considering that Lin-negative cells represent only 5% of GFP+ (ESC-derived) cells (Figure 1E), the overall number HSCs in the bone marrow is still low. The long time of FACS sorting can explain our technical difficulties with the requested experiments.

The authors did not have to test the average of GFP+ HSCs per chimaera, they simply had to show that some of the GFP+ cells were capable of cell autonomous long-term reconstitution. This brings to question the absolute number of mouse GFP+ HSCs in the chimaera bone marrow. Which should be determined from their flow cytometric analysis.

As requested by the Reviewer, we provided new Figure 1I which shows the absolute number of mouse GFP+ HSCs in the chimeric bone marrow determined by flow cytometric analysis. These new data were integrated in the Results section of the revised manuscript (pages 6-7).

The authors have to explain how they purified the ST- and LT-HSCs from the bone marrow and why 198 could not provide them with a viable recipient mouse when work from the Weissman and Morrison labs have shown that as few as 10 LT-HSCs (SLAM LSKs) can provide functional hematopoiesis in a transplant setting. Did the authors support the recipient mice with lineage+ splenocytes? What was the dose of irradiation administered? There has to be a better description of this experiment in the methods, in fact the methods as listed in the revision are very light on details of transplantation and lineage analysis.

We don’t doubt the high-quality experimental data from the Weissman and Morrison labs. As we explained above, this experiment didn’t work in our experimental settings due to, most likely, technical reasons which can include the long time of FACS sorting and/or insufficient support from lineage+ splenocytes among other reasons.

We would also like to point out, that the overall goal of our study was to produce a functional bone marrow (as an organ) from mouse ESCs in a rat, which was achieved using blastocyst complementation with subsequent analysis of the bone marrow by Flow cytometry, single cell RNA sequencing, and primary and secondary bone marrow transplantations into lethally irradiated mice. Testing functionality of every hematopoietic and stromal cell type in the bone marrow, even such important cells as HSCs, was not in the scope of this study but will be a subject of future research.

Because this experiment didn’t work, we don’t think that description of this experiment is needed in the Materials and methods section. However, we have added additional details of bone marrow transplantation and lineage analysis to the Methods section (pages 18 and 20) and discussed the limitation of our study about HSC functionality in the Discussion section (page 16).

So while the functional potential of chimeric stem cells has been established from the bulk bone marrow transplants, the lack of purified GFP+ mouse HSCs transplants limits the conclusions that the authors make with regard to calling these mouse HSCs functional in their experimental setting.

We understand the Reviewer concern that such experiment is needed to establish the functionality of HSCs. Therefore, we modified our manuscript to avoid statements that “the chimeric HSCs are fully functional” in our experimental setting. Instead, we used the words “the bone marrow from mouse-rat chimeras is functional”. This statement is fully supported by our studies because (1) the bone marrow from mouse-rat chimers rescued lethally irradiated mice and restored all hematopoietic cell lineages in the peripheral blood and bone marrow, including HSCs, after the primary and secondary transplantation; and (2) the bone marrow from mouse-rat chimers contained all ESC-derived hematopoietic progenitors and stromal cell subsets that were detected in normal mice and rats under our experimental conditions using the single cell RNA sequencing method.

Furthermore, we modified the Discussion section to acknowledge a limitation of our study about HSC functionality as follows (page 16):

“One of limitations of our studies is that the functional potential of chimeric HSCs was established from the whole BM transplants but not from transplantation of purified HSCs. While these experiments are technically challenging, transplantation of FACS-sorted donor HSCs into lethally irradiated mice will be needed in our future studies to investigate whether chimeric HSCs are fully functional to restore all hematopoietic cell lineages after irradiation.”

Reviewer #3 (Recommendations for the authors):This reviewer thanks the authors for the revised version of their manuscript that shows significant improvement.On the basis of this revised version, you will find enclosed my main comments for manuscript improvement. In general, I have the feeling that some of the figures are redundant or that the authors have trouble making a choice on what they want to show.

We would like to thank the Reviewer for additional comments and valuable suggestions for improvements of our revised manuscript.

Figure 1: The authors examine the percentages of the ST-HSC and LT-HSC in WT mice vs chimeric animals. Ideally, a mouse/mouse chimeric control should also be included. In the Sca-Kit analysis (Fig1E); the authors mentioned a significant increase in the Sca-Kit compartment in chimeras vs WT animals. This is substantiated in Fig1G, although two groups of animals with distinct (high and low) percentages of Sca-Kit are visible but not substantiated in Figure 1E where the authors show close percentages between the two conditions. The same remark applies in the opposite direction for the percentages of LT vs ST HSC. I would recommend more carefully choosing the panels to illustrate their conclusions. The rationale of analyzing the animals at P10 is rather uncommon in hematology unless you want to analyze the forming bone marrow. P30 would have been more classical.

We apologize for the lack of clarity in some Figure 1E panels. As suggested by the Reviewer, we revised our manuscript to replace Flow cytometry panels of the Sca-Kit analysis (Figure 1E). New Figure 1E panels are consistent with quantification of the data (Figure 1G-I). We would like to point out that our manuscript has a developmental focus since we analyzed embryos (E11-15) and postnatal time points (P5 and P10), providing a rationale for using P10 chimeras instead of more classical P30 time-point in our studies. The mouse control was provided in Figure 1E-I to show percentages of bone marrow cells in comparison with mouse-rat chimeras of same age.

Figure 2: The design of this experiment is still puzzling for me and is not clearly indicated either in the text or in the legend. The authors have sorted GFP+ cells from the BM. Why is it necessary to enrich in committed hematopoietic progenitors since the BM already contains high numbers of these cells? Please clarify.

We agree. We provided additional rationale for single cell RNAseq experiment in the Results section (page 7) and the Methods section (pages 18-19). Since the numbers of HSCs and other hematopoietic progenitor cells in the bone marrow are significantly low compared to numbers of differentiated hematopoietic cells, we enriched for BM progenitor cell populations prior to single cell RNA sequencing by combining 90% of FACS-sorted GFP^+^Lin^–^ cells and 10% of GFP^+^Lin^+^ cells in each experimental group. This enrichment enabled us to obtain enough progenitor cells for UMAP clustering analysis.

The rationale of mixing mouse/mouse and mouse/rat GFP+ cells is not clear since the authors do not identify cells originating from one combination vs the other.

We apologize for the confusion. We performed single cell RNAseq experiments separately for mouse-mouse and mouse-rat GFP+ cells. We made it clear in the revised Methods section (pages 18-19). We also removed old Figure 2A which created this confusion because in this figure both cell types were combined to show that mouse-mouse cells were indistinguishable from mouse-rat cells based on gene expression signatures. Revised Figure 2A show bioinformatic analysis of GFP+ cells separately for mouse-mouse and mouse-rat chimeras.

At this point, instead of performing a scRNA-seq, a GFP+ FACS sorting followed by a multilineage hematopoietic analysis by flow cytometry i.e.myeloid, lymphoid, erythroid cells would have been more simple and direct. Could you please clarify the design of the experiment?

We provided multi-lineage hematopoietic analysis of the bone marrow for Lin-negative, LSKs, ST-HSCs and LT-HSCs in Figure 1E-I and described it in the revised Methods section (page 20). The advantage of using single cell RNAseq is that many more cell types can be identified compared to FACS analysis. The design of scRNAseq experiment in Figure 2 was clarified in the revised Results section (page 7) and in the Methods section (pages 18-19).

More generally the authors strongly emphasize on the transcriptome analysis. Several main figures are redundant and could be switched to supplementary ones e.g. Figure 3 and Figure 4 since these latter does not bring any specific insights into species-specific mechanisms.

As the Reviewer suggested, we moved old Figure 3 into Supplemental material (new Figure 2 —figure supplement 5). Based on our data presentation, we cannot move old Figure 4 (new Figure 3) into supplemental material because this figure contains new single cell RNAseq experiment which shows both hematopoietic and stromal cells in the bone barrow. This experiment was requested in the first revision and provides an important support for the next main figure 4 and multiple supplemental figures that bring specific insights into species-specific mechanisms.

Fig6 should be revised. The sections of the embryonic aorta are not informative regarding the endothelium and the hematopoietic cell production. Please provide images showing low magnification of the aorta and the surrounding structures and high magnification of the hematopoietic clusters.

We agree. As requested by the Reviewer, we provided additional images showing low magnification of the dorsal aorta and the surrounding structures (new Figure 5A). High magnification of hemogenic endothelium and hematopoietic clusters is provided in Figure 5B.

Fig8: Multilineage analysis of the blood is interesting but the most informative piece of data comes from the BM analysis. Showing results at 8 days post-irradiation is not very informative since the animals are in aplasia.

We provided multilineage analyses of both BM and blood in main Figure 7 (old Figure 8) and six supplements to the Figure 7 (Figure 7 —figure supplements 1-6). The data were arranged to provide the most important findings in main Figure 7. The rest of the data were moved to supplements. Since we already obtained results at 8 days post-irradiation, we believe it is important to provide the readers this information and help them evaluating the aplasia and rescue experiments.